# Crystal plasticity as an indicator of the viscous-brittle transition in magmas

J.E. Kendrick [1], Y. Lavallée [1], E. Mariani [1], D.B. Dingwell [2], J. Wheeler [1] & N.R. Varley [3]

Understanding the flow of multi-phase (melt, crystals and bubbles) magmas is of great importance for interpreting eruption dynamics. Here we report the first observation of crystal plasticity, identified using electron backscatter diffraction, in plagioclase in andesite dome lavas from Volcán de Colima, Mexico. The same lavas, deformed experimentally at volcanic conduit temperature and load conditions, exhibit a further, systematic plastic response in the crystalline fraction, observable as a lattice misorientation. At higher stress, and higher crystal fraction, the amount of strain accommodated by crystal plasticity is larger. Crystal plastic distortion is highest in the intact segments of broken crystals, which have exceeded their plastic limit. We infer that crystal plasticity precludes failure and can punctuate the viscous-brittle transition in crystal-bearing magmas at certain shallow magmatic conditions. Since crystal plasticity varies systematically with imposed conditions, this raises the possibility that it may be used as a strain marker in well-constrained systems.

[1] Department of Earth and Ocean Sciences, University of Liverpool, Liverpool L69 3BX, UK. [2] Department für Geo- und Umweltwissenschaften, Ludwig-Maximilians-Universität, München, 80333, Germany. [3] Facultad de Ciencias, Universidad de Colima, Colima, 28045, Mexico. Correspondence and requests for materials should be addressed to J.E.K. (email: Jackie.kendrick@liverpool.ac.uk)

An understanding of the behaviour of magma at volcanic temperature and load conditions is fundamental to any inferences made regarding volcanic hazards. Towards this end, the rheological behaviour of magma during ascent must be characterised and its influence on on eruption style assessed. Laboratory experimentation can provide constraints on such processes by varying controlled conditions systematically[1, 2].

The silicate liquid state may be parameterised as a visco-elastic Maxwell body[3] which, acting as a Newtonian fluid at low strain rates, transitions to a non-Newtonian behaviour as deformation approaches the timescale of structural relaxation. The exsolution of gas bubbles[4–9] and nucleation of crystals[1, 10–19] yield further complexities in the rheological behaviour of multi-phase magmas[20–22]. Experimental studies using analogues[13, 14], synthesised materials[16, 22–27], natural variably glassy and crystalline magmas[15, 21, 27, 28] and partially re-melted intrusive igneous rocks—all demonstrate partitioning of suspensions during flow and deformation, as seen in many natural examples[29, 30]. These studies indicate that crystals interact during flow[20, 23, 29, 31], at or above a crystal content, which depends on the size and shape modality of the suspension[13, 14], which will also control the critical maximum packing fraction[11]. Crystal interaction results in strain-partitioning and shear-thinning behaviour of the suspension[16, 20, 25, 31–34], which in ascending dome lavas, favours the formation of discrete shear zones as strain localises along conduit margins[15, 32, 35–38]. This process can be key to the morphological development of an ongoing eruption, leading to plug flow that allows the bulk of the magma to ascend in a relatively undeformed state[32, 36, 39, 40]. The development of a viscous magma plug, essential for the exogenous growth of lava domes, can also be highly hazardous as gas pockets may become trapped below the relatively impermeable mass, creating overpressures that can destabilise the dome and lead to catastrophic eruptions[36, 41–43]. To unravel the propensity for localisation of strain in magmas at shallow crustal depths, here we examine magmatic textures and study their formation experimentally.

We demonstrate, for the first time in extrusive magma, that strain can also be accommodated by plastic deformation of the crystals, instead of wholly partitioned as viscous strain in the melt between a network of rigid suspended particles, that are susceptible to brittle failure. Crystal-plastic deformation occurs when the critical resolved shear stress on favourably oriented lattice planes is exceeded[44], resulting in dislocation movement and permanent strain. This deformation can be investigated using electron backscatter diffraction (EBSD), which measures crystallographic orientations[45–47]. Dislocations stranded in crystals may give rise to lattice distortion, a variation in crystal orientation within a single grain, indicative of crystal plasticity[48–52]. Crystal plasticity is not to be confused with visco-plastic (or Bingham) flow, often referred to in terms of non-Newtonian rheology in magmas[16, 53–56]. Although crystal plasticity has been observed many times in plutonic/batholithic structures[57–59], and inferred during shallow magma propagation[12], the role of plastically deformable particles on magma rheology has been considered negligible[16, 60, 61] or ruled out, and as such has been largely overlooked to date. Indeed to the best of our knowledge, no evidence of syn-emplacement crystal plasticity in silicic magma has been observed, and crystal plasticity has not previously been quantitatively documented in extrusive silicic magma. The quantitative data presented here is the first direct evidence of crystal plastic deformation in the crystals of both natural and experimentally deformed, erupted magmas.

## Results

**Characterisation.** Two blocks (COLB2 and COLLAH4) of natural andesitic lavas from Volcán de Colima (Mexico) were chosen for this study. The sample blocks, collected from a lava flow and a riverbed deposit, respectively, in 2004 (Supplementary Table 1), have near-identical chemical compositions (Table 1) and a phenocryst assemblage consisting predominantly of plagioclase (andesine to labradorite)[62, 63], with orthopyroxene and clinopyroxene, iron-titanium oxide and very minor amphibole set in rhyolitic glass with microlites of plagioclase and pyroxene, similar to other andesites resulting from recent activity at Volcán de Colima[64]. COLB2 has significantly lower porosity (9.5 vs 27.2%) and slightly higher crystal content than COLLAH4 (Table 2).

**Experimentation.** Each sample block was cored to prepare three cylinders of 25 mm (diameter) by 50 mm (height). The cylindrical cores of the magma were then deformed at 945 °C uniaxially under constant compressive stresses of 16 or 28 MPa to a total strain of 20 or 30% (Table 3). This yielded strain rates of $10^{-4}$ to $10^{-2.5}$ $s^{-1}$ and apparent viscosities of $10^{10}$–$10^{9}$ Pa.s (for further experimental detail, see Methods section and Kendrick et al.[32]) and provided three distinct sets of conditions (Supplementary Fig. 1), which could be compared to the starting materials. In addition, the samples chosen allowed examination of the effect of porosity and crystallinity on rheology.

The laboratory deformation experiments were performed at the pivotal change-point of the viscous-brittle transition (see Kendrick et al.[32] for more detail) and hence also led to significant amounts of brittle damage, particularly fractures in the phenocrysts of these multi-phase suspensions (Fig. 2 and Supplementary Figs. 2–6). The physical development of the samples included evolution of the porosity into sheared damage zones due to: growth and coalescence of pre-existing fractures; closure of pre-existing fractures perpendicular to the principal stress direction; and creation of new fractures parallel to the compression direction (Fig. 2 and Supplementary Figs. 2 and 3). This also led to a net decrease in permeability parallel to the principal stress direction[32], though through examining the thin sections (see Fig. 2 and Supplementary Fig. 2) this latter finding likely represents the anisotropic evolution of the material (observed in previous studies[65]) including denser regions adjacent to the ends of the samples and localised damage (and hence growth of localised permeable pathways) through the centre.

**Development of crystal plasticity.** Thin sections of the starting materials and six experimentally deformed cores cut parallel to the principal stress direction were analysed using electron backscatter diffraction (EBSD) to measure absolute crystallographic

---

**Table 1 Chemical composition of the starting materials**

| Sample name | Chemical composition (Weight %) | | | | | | | | | | |
|---|---|---|---|---|---|---|---|---|---|---|---|
| | $SiO_2$ | $Al_2O_3$ | $Fe_2O_3$ | MnO | MgO | CaO | $Na_2O$ | $K_2O$ | $TiO_2$ | $P_2O_5$ | Total |
| COLB2 | 60.89 | 17.93 | 5.72 | 0.10 | 2.82 | 5.81 | 4.76 | 1.34 | 0.60 | 0.20 | 100.49 |
| COLLAH4 | 59.10 | 17.45 | 6.13 | 0.11 | 4.04 | 6.62 | 4.52 | 1.19 | 0.62 | 0.19 | 99.97 |

Composition acquired by x-ray fluorescence, after Kendrick et al.[32]

orientations of crystals (Methods). In detail, this approach can identify lattice distortion within crystals, caused by dislocations stranded in crystals (Fig. 1). Since plagioclase comprises the largest proportion of both phenocrysts and microlites (Fig. 2), we limit our analysis to this phase, though we also indexed some pyroxenes (Supplementary Figs. 4 and 5). We found the deformation of phenocrysts to be dominated by well-defined fractures (Supplementary Fig. 6), which often formed parallel to the principal stress direction (Fig. 2 and Supplementary Fig. 3), suggestive of the development of a rigid framework of phenocrysts in which stress built-up within the crystals[12] until their strength was exceeded. Previous experimental deformation of multi-phase suspensions has highlighted the tortuous temporal evolution of fracture damage through melt and crystals[15, 36, 66, 67], while the complexity of natural magmatic shear zones may be obscured by healing due to the relaxation of the melt phase[3, 68, 69]. Here, fractures are evident in the experimentally deformed samples (Fig. 2), and this fracturing resulted in a net grain size reduction during deformation of both the phenocrysts (visible in Supplementary Figs. 4 and 5) and microlites (quantified in Supplementary Fig. 7). As our focus here is plasticity, and to provide a more statistically relevant data set, we focus our further analysis on microlites, allowing us to analyse >50 crystals from each sample (Fig. 3 and Supplementary Figs. 8 and 9). These microlites are subject to small local fluctuations in the crystallographic preferred orientation (CPO) around larger heterogeneities such as pores and phenocrysts (Fig. 3), but no systematic CPO across samples that would indicate pre-existing strain localisation. Importantly, we demonstrate that the andesitic magma samples, first deformed naturally during eruption and then experimentally in the laboratory, exhibit crystal plasticity in the microlites.

Plasticity is identified by distortion of the crystal lattice, quantified as a 'misorientation', which can be seen in the colour-graded images produced with respect to a chosen reference point within individual crystals (eg, Fig. 4a). These misorientations result from crystal lattice rotations produced by the movement and accumulation of dislocations[50] in the plagioclase crystals (Fig. 1). The distortion data from transects across individual crystals are summed into misorientation profiles (Fig. 4; the full data set is available in the Supplementary Data), while pole figures

allow us to explore this distortion in more detail (Fig. 4c), further highlighting even subtle distortions from a 'pristine' microlite structure. The distortion within the microlites is a function of the strain field seen by each of them individually. Some show continuous and increasing distortion (Fig. 4b), while others display bending whereby misorientation with respect to the reference point increases and then decreases (Fig. 4f).

In the experimentally deformed samples, some microlites are also broken into distinguishable fragments or contain fractures (Fig. 4e; Supplementary Figs. 8 and 9). These crystals are characterised by a gradual lattice distortion (by dislocation), punctuated by an abrupt jump in misorientation across the fracture (Fig. 4f). Broken fragments may be characterised by further rotations after rupture that passively misorient them with respect to the parent microlite (the degree of misorientation across the fracture shown in Fig. 4f has no significance in terms of crystal plasticity). Plasticity and fracturing are common deformation mechanisms in plagioclase, where fracturing is observed to occur simultaneously with, or as the culmination of, crystal plastic deformation[70]. Irrespective of sample, we find that plagioclase microlites are elongated preferentially parallel to their *a* axis (as is often observed in magmatic plagioclase[71]; see Supplementary Fig. 10), and no sub-grain boundaries are seen in any of the microlites studied. Previous work on plagioclase found dislocations moved on up to four slip systems[72], though predominantly on (010) [001][72–74], where the plane (listed first) is the slip plane and the direction is the Burgers vector. Transmission electron microscopy (TEM) and crystallographic studies on plagioclases deformed at high temperature suggest screw dislocations on (010) [001] dominate, with slip on (010) [100] of secondary importance[75]. Stunitz et al.[70] showed that in experiments (001) [110] and (010) [001] slip systems were equally common, with (111) [110], ($\bar{1}$31) [101] and ($\bar{2}$42) [101] also possible. Distortion can be quantified by a misorientation gradient, but can be analysed in more detail to give information on the Burgers vectors of the stranded dislocations. To investigate this, several crystals were chosen for further analysis of the slip systems using the weighted Burgers vector (WBV) method devised by Wheeler et al.[76] This method provides a quantification of the density of dislocations with particular Burgers vectors, thus supporting the inference of most common slip systems (the WBV technique is described in Methods). The WBV can be calculated at each point, and its direction plotted in an inverse pole figure to make an easy link to crystallography. Alternatively, it can be averaged over an area and given as a vector **K** resolved into components in $(\mu m)^{-2}$ parallel to crystallographic directions *a*, *b* and *c*. Although biased or 'weighted', it cannot generate phantom Burgers vectors. The average WBV in rectangular regions in a subset of the microlites is characterised by high *c* values with variably lower *a* and *b* (Fig. 4). Local WBV directions (Fig. 4) show a maximum near [001], though with some spread, mainly towards the *b* axis, characteristic of the slip systems most commonly observed in plagioclase.

**Table 2 Physical attributes of the starting materials**

| Sample name | Pores (%) | Solid portion (%) | | |
| --- | --- | --- | --- | --- |
| | | Glass | Phenocrysts | Microlites |
| COLB2 | 9.5 ± 1.0 | 28.7 | 37.6 | 34.2 |
| COLLAH4 | 27.2 ± 0.8 | 41.2 | 27.5 | 31.6 |

Porosity measured on five cores from each sample set (COLB2 and COLLAH4) with the range indicated. The glass, phenocryst and microlite contents were measured from optical analysis of the starting materials (see Kendrick et al.[32]) and converted to amount (%) of the solid portion

**Table 3 Experimental conditions for uniaxial compressive tests**

| Sample name | Stress (MPa) | Strain (%) | Time (s) | Mean strain rate (s⁻¹) | Mean temperature (°C) | Log mean viscosity (Pa.s) |
| --- | --- | --- | --- | --- | --- | --- |
| COLB2 | 15.8 | 27.1 | 3010 | $10^{-3.99}$ | 945.5 | 10.6 |
| COLB2 | 28.4 | 19.7 | 600 | $10^{-3.45}$ | 945.5 | 10.3 |
| COLB2 | 28.4 | 29.1 | 858 | $10^{-3.45}$ | 945.8 | 10.3 |
| COLLAH4 | 15.9 | 28.5 | 320 | $10^{-3.01}$ | 941.0 | 9.6 |
| COLLAH4 | 28.4 | 18.6 | 64 | $10^{-2.54}$ | 952.0 | 9.4 |
| COLLAH4 | 26.4 | 30.1 | 173 | $10^{-2.77}$ | 943.8 | 9.5 |

The rheological data is presented in detail in Kendrick et al.[32]

By compiling distortion data for all the crystals analysed in both the natural and experimentally deformed materials, it is possible to identify trends (Fig. 5). For all sample sets, the total misorientation of the crystal lattice tends to be higher in longer crystals (Fig. 5a–c) and those with higher aspect ratio (Supplementary Fig. 11). Applying the same technique to the broken crystals (from all experimental conditions) reveals the same trend of increasing misorientation with increasing crystal fragment length (Fig. 5a, b). By dividing the maximum misorientation by the long axis length of the crystal, we create a metric 'misorientation per micron' for each crystal measured, which allows for comparison of deformation intensity across microlites of different sizes. We find that the misorientation per micron is systematically higher in samples deformed at higher stress, that the maximum misorientation per micron is higher in the more deformed samples (higher strain), and that the broken crystals have the highest misorientations across all samples (Fig. 6). Using the mean misorientation per micron for each sample set, we can create a prediction tool that allows an estimate of the amount of distortion in a crystal of given length in any sample (creating the lines in Fig. 5). This allows us to compare the large data set (Supplementary Data 1–11) quantitatively, and demonstrates that length has a more systematic control on misorientation intensity than aspect ratio does (Supplementary Figs. 11 and 12). We identify that the experimental deformation of these magmas at high temperatures led to an increase in the amount of crystal distortion (larger misorientations) in the plagioclase microlites (Fig. 5). The interplay between stress and strain depicted here is complex. The mean, the minimum and the maximum values of misorientation per micron always increase as deformation conditions increase (from left to right in Fig. 6) in both samples, ie, as either stress or strain is increased. The median, 25th and 75th percentiles increase more systematically with stress, and interestingly, there is little change (in the median, 25th and 75th percentiles) from 20 to 30% strain at 28 MPa, suggesting some microlites see increasingly high values of plastic strain, resulting from the stress accumulation in shear zones, while others fail to deform more as strain increases. Previous work has demonstrated that, at higher stresses, strain localisation occurs earlier, resulting in brittle behaviour at lower total strains[32, 36]. So, while the deformation of some microlites certainly increases (the min., max. and mean increase across the same range), there could be an increasing prevalence of strain-partitioning as strain increases[16]. This subtlety is also observed in Fig. 5, where over the range investigated, increasing stress (green to blue from 16 to 28 MPa) has a more pronounced effect on lattice misorientation than increasing strain (red to blue from 20 to 30%). Increasing either stress or strain increased the proportion of fractured crystals, resulting in a net decrease in grain size, which is more pronounced in the sample with higher crystal fraction, COLB2 (see Fig. 5a, b, and Supplementary Fig. 7).

The fact that this grain size reduction leaves broken crystals with the highest recorded misorientations observed in this study (Fig. 5c), having more than twice the misorientation value (hence distortion) of microlites in the starting material for a given crystal length (Fig. 5c) may be key to understanding the plastic limit[77], and provide a threshold for, or lower-bound estimate of, the maximum lattice misorientation that a plagioclase microlite can accommodate before fracturing, considering that these high distortions (of up to 10°) are not seen in the intact, coherent crystals (Fig. 5). Indeed, in the starting materials and at each set of experimental deformation conditions, crystal distortions are

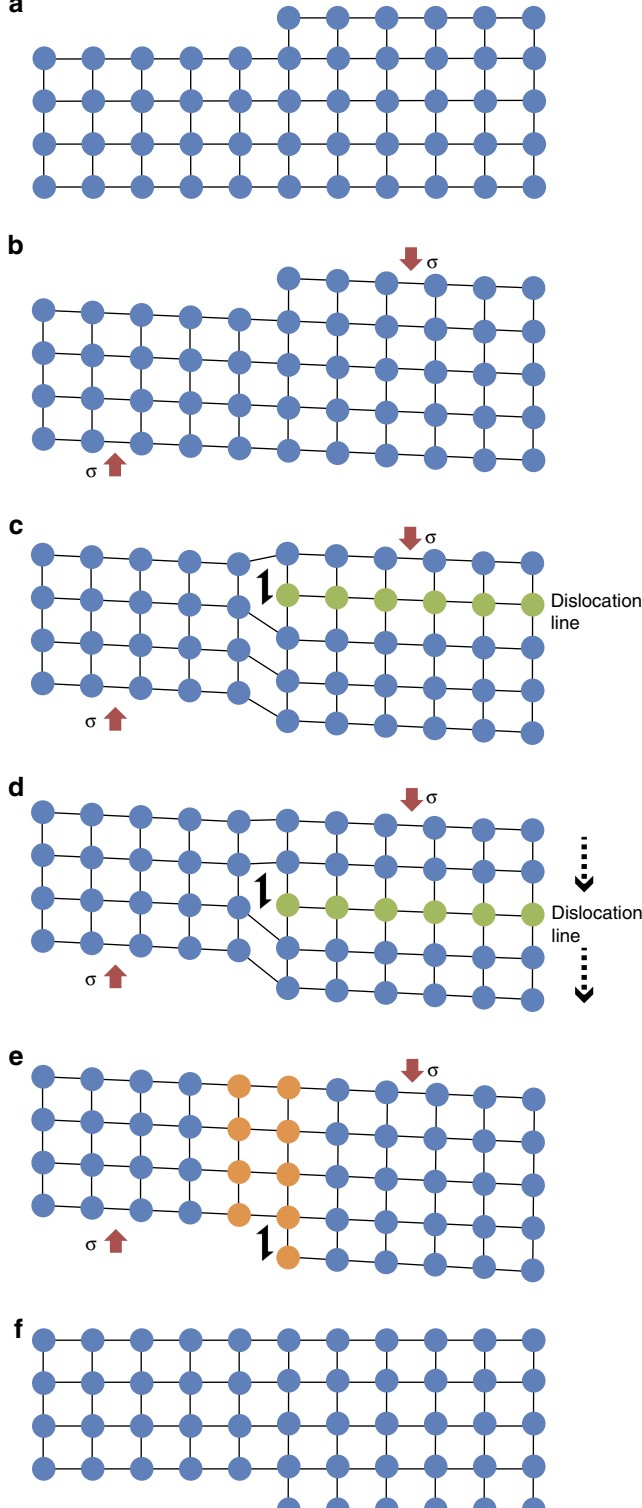

**Fig. 1** Schematic representation of a dislocation in a crystal lattice. Here an edge dislocation forms and propagates through the lattice for simplicity, but screw dislocations are also common. The stages show: **a** isostatic stress conditions, and no deformation; **b** differential stress applied, leading to elastic strain; **c** yielding occurs under applied stress, creating a dislocation; **d** the dislocation migrates under differential stress conditions; **e** the dislocation passes through, resulting in distortion and **f** upon return to isostatic stress conditions, elastic strain is recovered. Dislocations that have passed through as in **f** leave a shape change but no internal distortion, however, significant densities of dislocations stranded within the lattice at stages **c–d** give rise to macroscopic lattice distortion as documented here

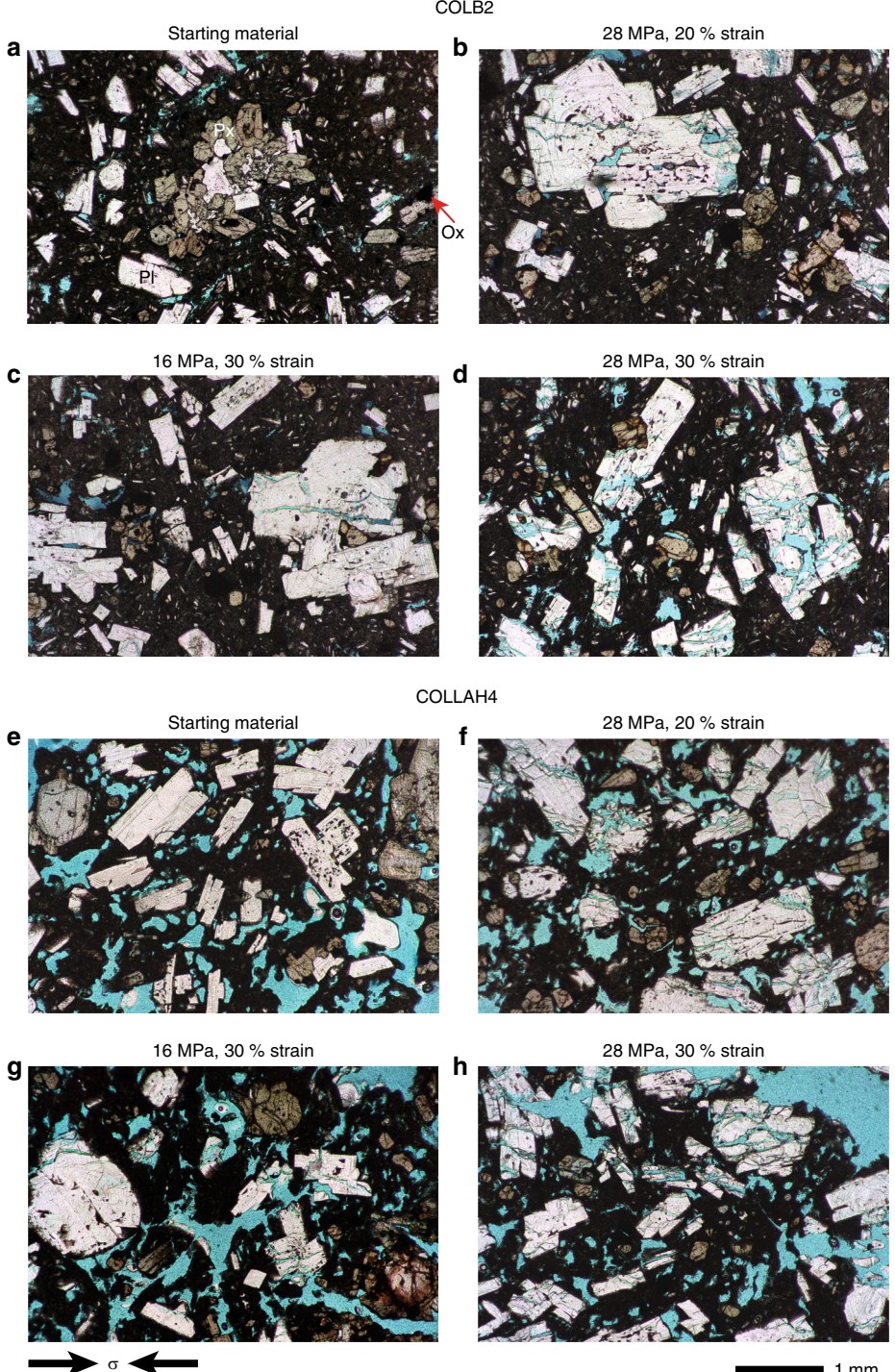

**Fig. 2** Photomicrographs of the natural and experimentally deformed samples. COLB2 has relatively few pores (in blue), and those present are small and intermittently clustered in the starting material (**a**). COLLAH4 starting material has more, larger, vesicle-shaped pores (**e**). In both samples (**b**–**d** for COLB2, **f**–**h** for COLLAH4), applying stress and strain resulted in fracturing sub-parallel to the compression direction (compression direction is horizontal in the current view). These fractures are first seen in the crystals at lower stress or strain conditions, and then as imposed stress or strain is increased, the fractures coalesce through the interstitial glass, and grain crushing dominates. Scale bar is 1 mm and refers to all panels. Pl plagioclase, Px pyroxene, Ox iron-titanium oxides

higher in the lower porosity, more crystalline sample COLB2 (Figs. 5 and 6), due to the higher likelihood of crystal–crystal interactions. However, fracture coalescence and the development of a pervasive damage zone appears more easily attainable in the initially more porous COLLAH4 (Fig. 2 and Supplementary Figs. 2 and 3).

## Discussion

Using two different partially crystalline magmatic suspensions with some initial crystal plasticity, and inducing variable amounts of strain at different applied stresses, we find that crystal plastic deformation of plagioclase is attainable under shallow magmatic temperature and stress conditions. Crystal distortions observed in

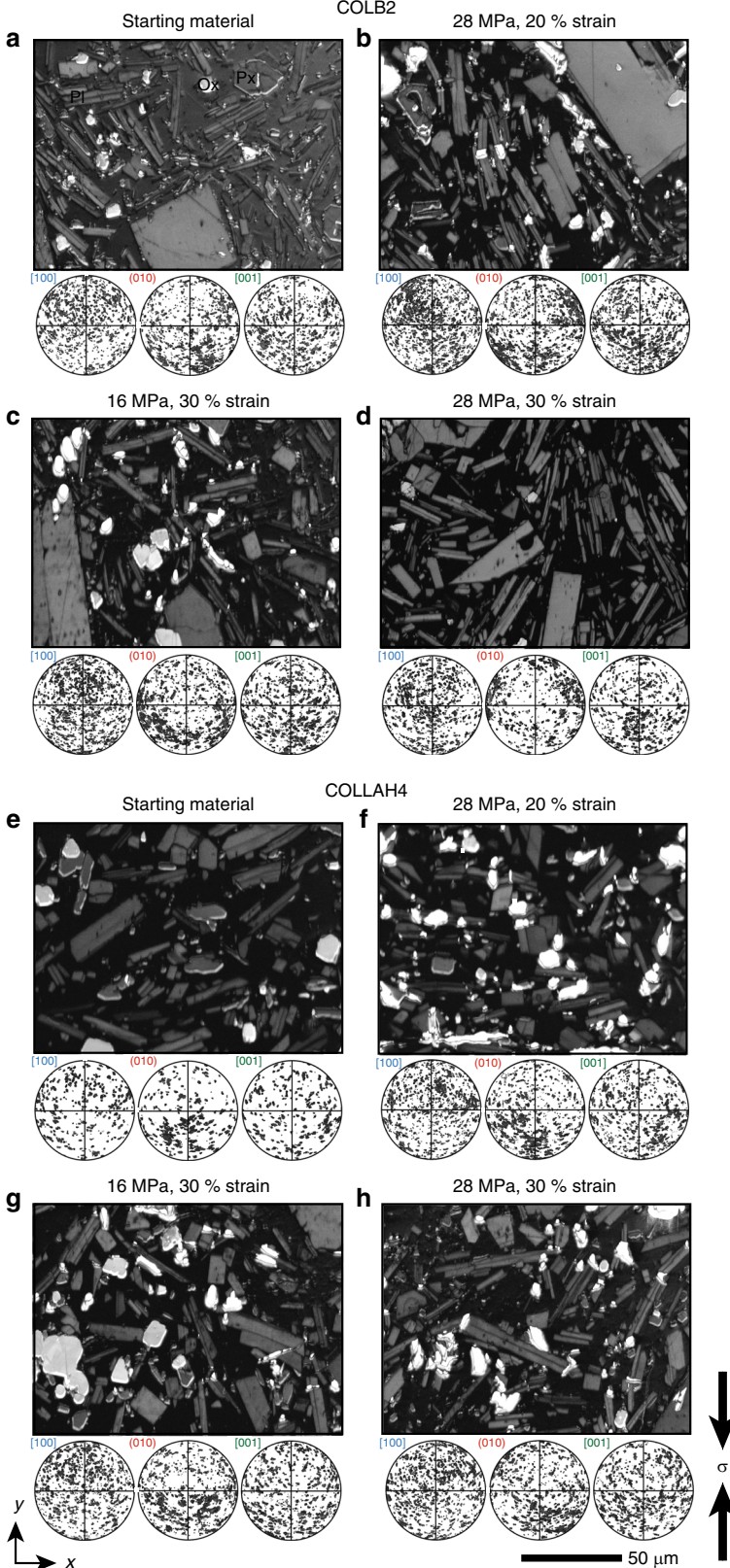

**Fig. 3** Crystallographic orientation in natural and experimentally deformed andesites. Band contrast images from EBSD mapping, showing one mapped area per sample and pole figures [100] (010) [001] (lower hemisphere) for all microlites measured for the starting material (COLB2 in **a** and COLLAH4 in **e**) and experimentally deformed cores (in **b**–**d** and **f**–**h** for COLB2 and COLLAH4, respectively). Scale bar is 50 μm and refers to all panels. In the pole figures, local fluctuations in crystallographic preferred orientation (CPO) are observed, but there is no apparent systematic CPO across the sample sets (all cores are cut in the same orientation, with the principal stress direction in the experimentally deformed samples vertical here). Pl plagioclase, Px pyroxene, Ox iron-titanium oxides

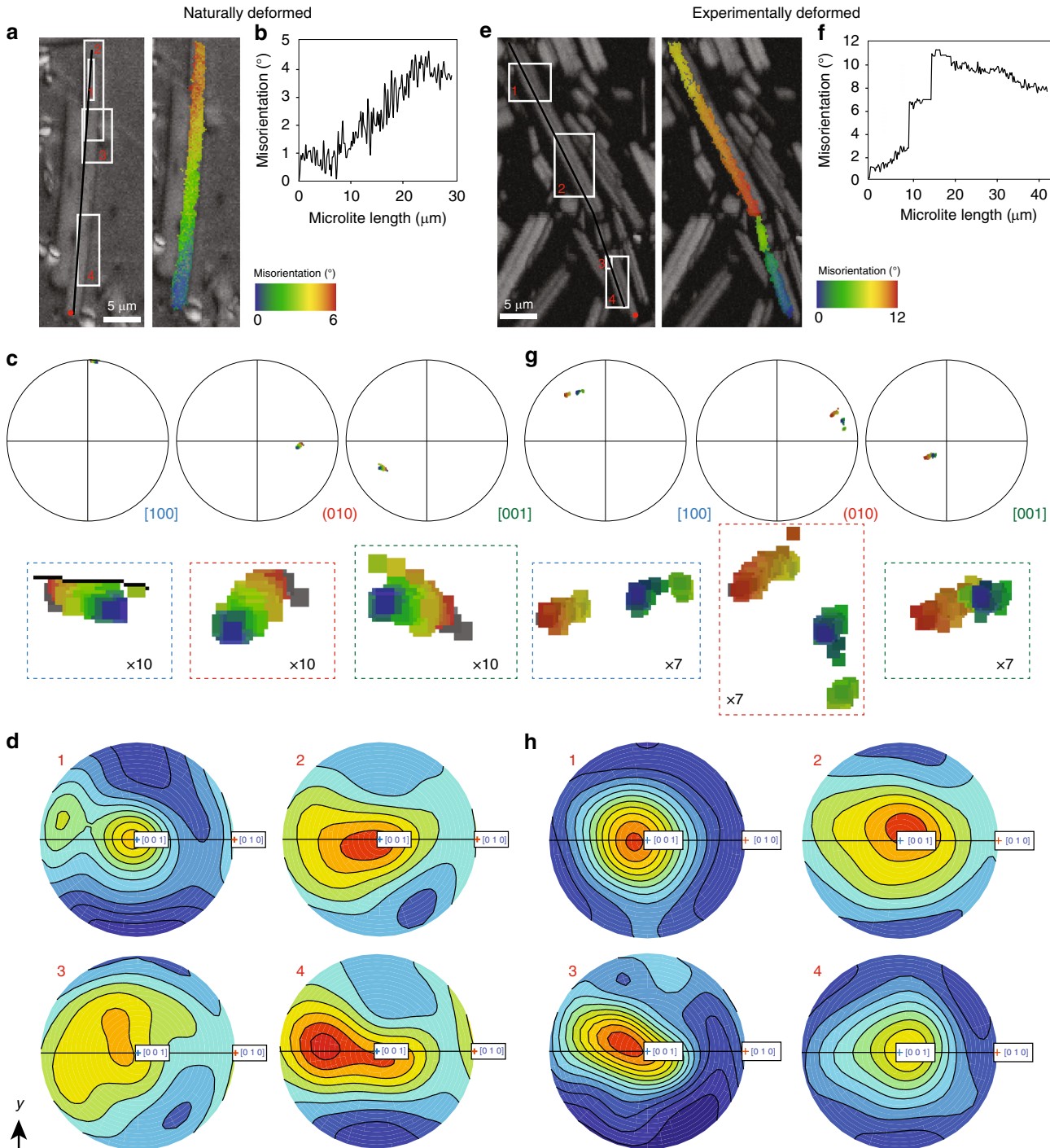

**Fig. 4** Crystal-plasticity in plagioclase. **a** Band contrast image with location of WBV analysis areas 1–4 misorientation transect line and texture component map showing crystallographic misorientation in a plagioclase microlite (example from COLB2 starting material)—colour varies due to distortion in the crystal lattice from a reference point, according to the scale given (here 6°), with progressive rotation indicating crystal plasticity resulting from dislocation; **b** the misorientation profile along the black line on the crystal in **a** (from the orange dot in the image at 0,0 on the plot), showing > 4° misorientation across the crystal; **c** pole figures [100] (010) [001] (lower hemisphere) show the deformation of the same microlite (with the same colouring as in **a**) partitioned in each crystallographic axis. **d** Slip systems using the weighted Burgers vector (WBV) method devised by Wheeler et al.[76], which shows the Burgers vectors of dislocations of the areas marked 1–4 in **a**. The **K** vector components in (µm)$^{-2}$ for box 1 in **a** are: $a = 6.21$, $b = -1.83$ and $c = 16.65$; **e** as **a**, for a broken microlite (example from COLB2 28 MPa, 30% strain sample) showing crystal plasticity within the three distinct segments, with the compression direction vertical in the image; **f** as **b** for the same crystal in **e**—the misorientation within the intact segments is gradual, but is punctuated by abrupt misorientation increases across the fractures in the microlites; **g** pole figures [100] (010) [001] (lower hemisphere) show the deformation of the microlite in **e** (with the same colouring), with each of the three fragments plotting as isolated patches that spread along the same axes, highlighting that the fragments have been displaced within the sample following fracture; **h** slip systems of boxes 1–4 in **e**, using the WBV method, which shows the Burgers vectors of dislocations. The **K** vector components in (µm)$^{-2}$ for box 3 in **e** are: $a = 0.13$, $b = -7.10$ and $c = -21.17$

microlite sections elongated parallel to the *a* crystallographic axis are most likely caused by slip on the (010) [001] slip system, commonly observed to be a 'soft' orientation in plagioclase[72–74]. We propose here that screw (or mixed character) dislocations on (010) [001] are most likely to cause the rotations of up to 10° observed in these sections, as confirmed by the WBV data. We cannot rule out the role of slip on (001) and (111) planes along

the <110> direction[70], which may also contribute to such lattice distortion. In addition, the lack of any subgrains in the microlites suggests that, in natural and experimentally deformed andesite, recovery mechanisms and rotation recrystallisation are not active. The lack of any recrystallisation has been observed previously in single crystal studies, at temperatures up to 900 °C[78].

Crystal-plastic deformation may serve as an important outlet for strain: it may be viewed as the mechanism that accommodates some of the permanent, inelastic deformation imparted on magma that does not reside in the indefinitely deformable liquid phase when approaching eruption. The susceptibility of plagioclase to crystal plasticity is of particular relevance to the study of silicic volcanoes, where the mineralogy is commonly dominated by this phase. As deformation increases, so the crystal plasticity increases, and, in the absence of recovery and with the accumulation of dislocations leading to considerable distortion, microlites exceed their plastic limit and suffer brittle fractures. This demonstrates that, in an ascending or extruding multi-phase magma, stress and strain during viscous flow can lead to crystal plastic deformation that eventually results in brittle failure, evidenced by brittle microstructures commonly preserved in the crystalline phase present in shear zones at lava domes[36, 79–82].

We conclude that the non-Newtonian rheology of multi-phase magma is controlled not only by the porosity, the packing fraction and the size, shape and dispersal of crystals[11, 20, 21, 31, 34, 66], but also by the deformability of the particles in suspension. The deformability of crystals has important repercussions: first, for our understanding of maximum packing fractions, a concept understood only for rigid particles with relatively simple size and shape modality[11]; second, for the compressibility of magma[83] and third, in the release of melt from partially molten or partially crystallised bodies in the Earth. Further, crystal plasticity necessarily complicates the viscous-brittle transition envisaged during magma ascent[19, 25, 32, 36, 84, 85], providing a time–space interval during which strain may be accommodated by crystal plastic deformation (Fig. 7). The locus of this crystal-plastic interval will be dependent upon total strain/strain rate (~ascent), total shear stress (which increases towards the margins of the conduit) and crystallinity/porosity (itself dependent upon composition and temperature) that will control local stress concentrations[12], strain partitioning and strain localisation[21, 23, 32]. For example, in the rheological (stress-strain-temperature) window examined here, only the plagioclase microlites deformed plastically, but under differing conditions plastic strain may become relevant in other phases and at other scales. For example, crystal plasticity has been noted in intrusive magmatic settings[57], and in experiments on

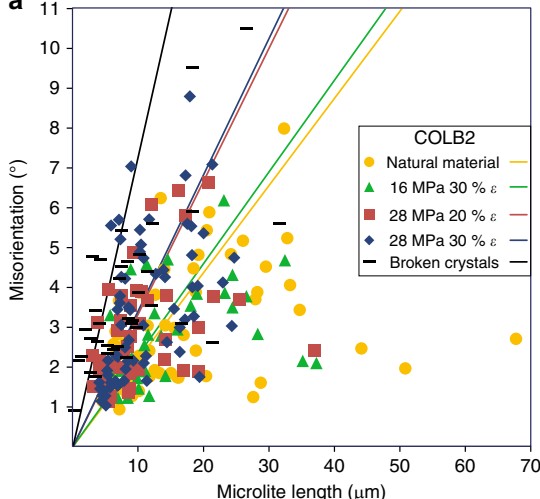

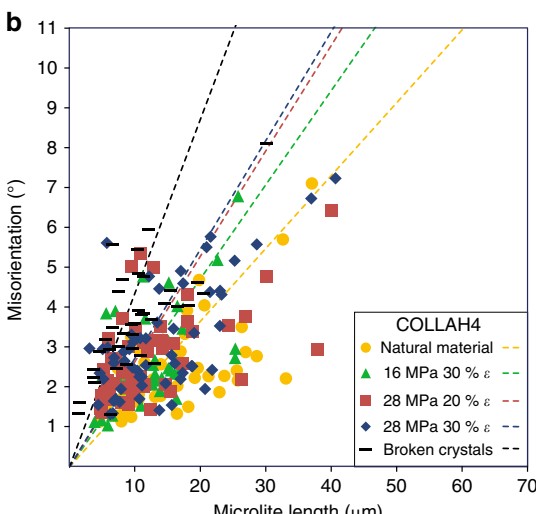

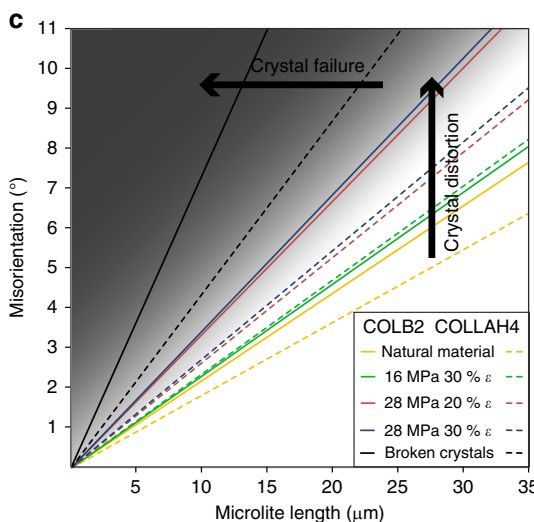

**Fig. 5** Quantified lattice distortion in microlites. Misorientation value versus crystal length for all the microlites analysed for **a** COLB2 and **b** COLLAH4 starting material, samples deformed at 16 MPa to 30% strain, 28 MPa and 20% strain, 28 MPa and 30% strain or broken crystals across all sample sets (only misorientation from plastic deformation were considered in these data). The points shown indicate an increase in misorientation and shortening length (grain size reduction) with increasing deformation, and correspond to the data in Supplementary Fig. 7 and Supplementary Data 1–11. The gradients of the lines depicted indicate the mean misorientation per micron for microlites in each given sample set (see also Fig. 6), highlighting the effect of increasing stress and strain on crystal plasticity. **c** The lines taken from **a** and **b** show how increasing stress or strain increases the measured value of misorientation for a given sample length, with the highest (crystal plastic) misorientation values in the broken crystals in both COLB2 and COLAH4, suggesting they may have reached and exceeded a plastic limit. It also shows higher deformation in the less porous COLB2, in both the starting materials and at every stage of sample deformation, suggesting a higher degree of interaction between microlites

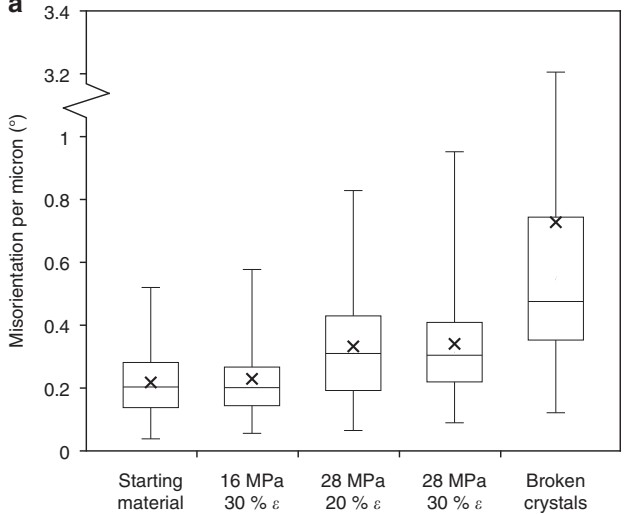

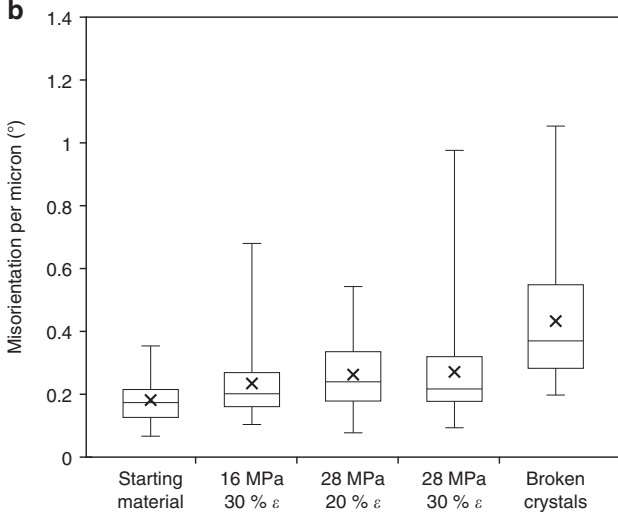

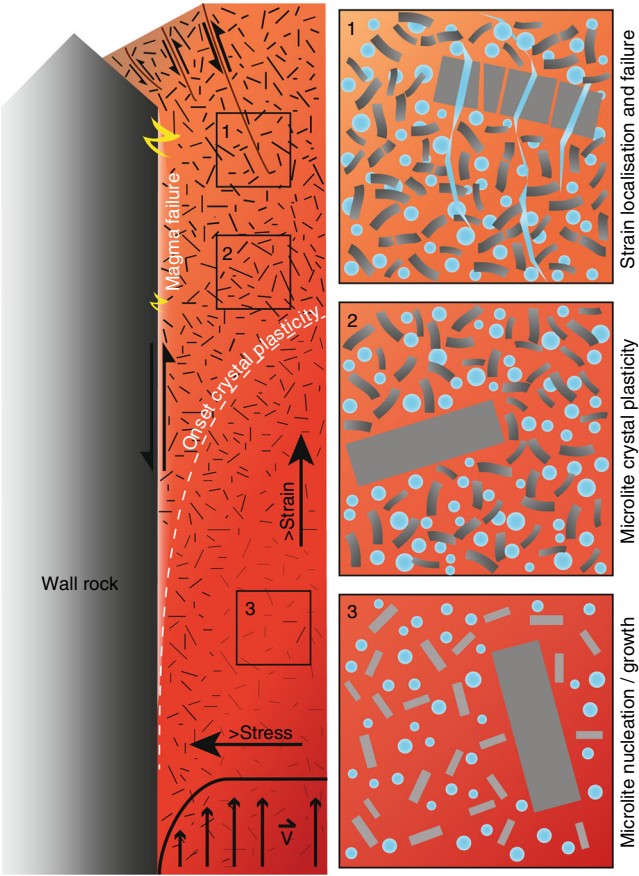

**Fig. 7** Deformation mechanisms in the conduit. Schematic of conduit processes and the locus of deformation mechanisms inferred in ascending magma, with panel insets (1–3) highlighting these processes at a finer scale. This shows the velocity profile active along the length of the conduit (V̄), and the transition from bulk viscous flow during crystallisation in the lower conduit (panel 3), through to strain partitioning that results in crystal plasticity in the microlites (indicated by grey scale variation in panel 2) in the mid-upper conduit and near the margins (where strain localises and shear stress is higher), and finally to the onset of brittle deformation with further increasing stress, strain or strain rate as the strength of the crystals and interstitial melt is exceeded and fractures propagate (panel 1) in the upper part of the conduit

**Fig. 6** Box-plot of deformation intensity across sample sets. The spread in misorientation per micron (simply the lattice distortion (in degrees) divided by the crystal length (in microns)) for **a** COLB2 and **b** COLLAH4. This shows the maximum, minimum (the top and bottom ends of the vertical lines), median (the central horizontal line), the 25th and 75th percentiles (the bottom and top of the boxes, respectively) and mean (crosses) of the values for the starting material, each stress and strain ($\varepsilon$) condition and the broken microlites. Using the misorientation per micron allows us to exclude any bias that would result from comparing microlites of different sizes

partially molten granite analogues, relevant to lower crustal environments, Mecklenburgh and Rutter[61] and Rutter et al.[60] measured stress exponents, $n$, between 1 and 2, and interpreted this as indicative of dominant diffusive mass transfer processes, while tentatively attributing the non-linear component to grain contact processes. These authors also suggest that the influence of intracrystalline plasticity on the rock rheology is insignificant. However, some undulose extinction and cracking in quartz grains are observed in their experiments[61], but attributed to preparation techniques. Quantitative analyses of the microstructures they observed would be needed to clarify the role of crystal plasticity in their experiments. The small strains accommodated by crystal plastic deformation, potentially seen in Mecklenburgh and Rutter[61], but certainly identified in this study and in numerical simulations[12], may not initially influence the overall rheology of the magma, however, the onset of crystal plasticity may be the

most important marker of the development of locally high stresses that subsequently lead to fracturing, it may therefore signal the moment in space and time when the viscous-brittle transition is approached by magma in a volcanic conduit. As such, the systematic variation in plasticity as a result of stress and strain conditions seen in our experiments, has significance for the interpretation of the deformation history of magmas. Because the silicate interstitial melts of magmas are visco-elastic fluids that are able to relax an applied stress[68], according to their relaxation timescale (which depends upon their viscosity) and hence heal and recover following deformation, the identification of strain markers[86] is key to unravelling their deformation histories. For example, bubbles in explosive volcanic products are used to establish deformation conditions immediately prior to fragmentation, since the structures quench as they are expelled[86, 87]. However to unravel the history of extrusive volcanic products, such as lava domes (which form over longer timescales), we must rely on other, as yet unidentified, indicators to differentiate stress and strain conditions, enhancing our knowledge of strain partitioning; a common and integral feature of these types of

eruption[82]. In these scenarios, plastically deformed crystals could be used to indicate strain localisation and magnitude; indeed with integration into rheological models, crystal-plastic deformation may well become a valuable tool that helps characterise magma transport and deformation processes across a wide range of magmatic settings.

## Methods

**Characterisation.** Bulk chemical composition was measured by X-ray fluorescence for the pristine starting materials and is presented as oxide weight % in Table 1. Connected porosity was measured via the Archimedes buoyancy method, results are presented in Table 2 along with the variability from measurements on five cores, and the glass, phenocryst and microlite content was converted to a solid fraction from image analysis in Kendrick et al.[32].

**Experimentation.** The cylindrical cores of 25 mm (diameter) by 50 mm (height) of the magma were placed in a 3-zone split-cylinder furnace around a uniaxial press and heated at 2 °C per min until stabilisation at 945 °C ± 7 °C (thermal equilibration measured using a thermocouple embedded into the centre of the sample). Deformation proceeded under constant uniaxial compressive stresses of 16 or 28 MPa (applied near-instantaneously) to a total strain of 20 or 30%. Displacement and load were measured at a rate of 10 measurements per s. Viscosity ($\eta_a$) was calculated using equation 1, the parallel plate method of Gent[88]:

$$\eta_a = \left(2\pi F h^5\right)/\left(3V(\mathrm{d}h/\mathrm{d}t)(2\pi h^3 + V)\right), \qquad (1)$$

where $F$ is force (N), $h$ is the length (m), $V$ is the initial volume of the sample ($m^3$) and $t$ is time (s). Thin sections were made of the starting material and six experimentally deformed cores (parallel to length/principal stress direction) and were analysed by EBSD to measure absolute crystallographic orientations of crystals.

**EBSD analysis.** EBSD maps are formed by arrays of data points (each carrying crystal orientation information) organised in a regular grid where the spacing between each point (step size) is decided by the operator and based on the nature of the study. EBSD was conducted in a CamScan × 500 CrystalProbe field-emission gun scanning electron microscope (SEM) and a tungsten filament Phillips XL30 SEM using 40 and 3 nA beam currents, respectively, and 20 kV accelerating voltage, spot size of 6 μm and working distance of 25 mm. The angle of incidence of the electron beam on the sample surface is 70° (see the section 'The relationship between EBSD and SEM reference frames' for details of the ×500 and XL30 SEM systems geometry). Maps were made of two or more areas in the centre of the thin section; one of > 4 × > 8 mm on the XL30 SEM and one or more smaller area of ~140 × 100 μm using the ×500 CrystalProbe SEM. Data acquisition was performed with the AZtec EBSD software from Oxford Instruments HKL using a step size of 15 μm and 0.2 μm for the large and small maps, respectively. Minerals were indexed using 12 bands, 70 reflectors, 120 Hough resolution, curved band edges and 2 × 2 binning, appropriate for indexing plagioclase and with an error on measurements of orientations of ±0.5°.

Data processing was carried out using the CHANNEL 5 software by Oxford Instruments HKL. EBSD maps presented here are: band contrast (BC), phase, all-Euler angle (AllE) and texture components (TC) maps. BC maps (as in Fig. 3) are routinely represented using a grey scale with assigned values from 0 (black) to 255 (white). These maps reflect the quality of the Kikuchi pattern indexed; sharp, good quality patterns return light pixels, poorly defined patterns return dark pixels. BC maps are used as the background for other data sets in Fig. 4 and Supplementary Figs. 4–6 and 8–10. Phase maps are colour-coded according to the mineral phases present, as identified by the EBSD software using the crystal structure files provided by the operator (phase maps are presented in Supplementary Figs. 4 and 5). In AllE maps, each colour represents a given crystallographic orientation, described by a combination of the three Euler angles (AllE maps are presented in Supplementary Figs. 4, 5 and 8, 9). AllE angles use RGB colour coding, whereby the combination of three Euler angles that takes us from the SEM reference frame to the orientation of crystal, is visualised as one final combined colour[89] (eg, purple). Subtle misorientations (< 5 degrees) within that crystal cannot be seen clearly using the all-Euler colour scheme (because large changes in absolute orientations are being measured), which is where the TC is introduced. TC maps are useful tools that assist with the analysis of crystallographic distortion. They are produced by identifying, and assigning a colour to, a reference orientation in the crystal of interest. The maximum distortion in the crystal (from the reference orientation) can then be measured and highlighted using a pre-defined colour range. Individual crystals were subjected to analysis of lattice distortion, where the degree of misorientation of the crystal lattice relative to a selected reference pixel at one end of each microlite is shown (TC maps are shown in Fig. 4 and Supplementary Fig. 6). Misorientation profiles were also made along length of all the crystals analysed (Supplementary Data 1–11) and can be plotted to give a graphical description of the deformation they experienced (eg, Fig. 4, Supplementary Fig. 6). These measured crystal lattice misorientations are considered to be a minimum

value, as it is unlikely that the principal strain axis of the crystal lies along the plane of the thin section surface. Finally, [001] (010) [100] pole figures (lower hemisphere) of the microlites were constructed to further examine crystal lattice distortions (rotations) due to dislocations in individual crystals, and to examine any CPOs.

The weighted Burgers vector (WBV) method devised by Wheeler et al.[76] provides information on the Burgers vectors of 'geometrically necessary' dislocations, ie, those which give rise to visible distortion. The method is biased towards dislocations whose lines are more steeply inclined to the EBSD map, hence the prefix 'weighted', but it cannot artificially generate Burgers vector directions that are not actually present—hence its usefulness. Local WBVs are calculated from the differences in orientation of small groups of pixels. Average WBVs over areas are expressed by the **K** vector resolved into components in (μm)$^{-2}$ parallel to crystallographic directions $a$, $b$ and $c$. Both calculations have errors because there are small errors in the measurements of orientation using EBSD. The local and average calculations are consistent but the **K** vectors are calculated by integration of crystal orientation around the perimeter of the selected region and error bars are consequently smaller[76]. We used both methods; currently the software can average only rectangular regions, which presented challenges due to the narrow morphology of the microlites, but local and average calculations are in broad agreement.

**The relationship between EBSD and SEM reference frames.** $X_s$, $Y_s$ and $Z_s$ are defined as the axes of the sample reference frame. A 180° rotation around the $Z_s$ axis (vertical in the Liverpool EBSD system as the sample is held horizontal) between the reference frame of the SEM image and that of the EBSD camera is inherent to all SEM-EBSD systems. Britton et al.[90] provide the mathematical framework necessary to establish a consistent convention that relates these two reference frames working from first principles. The Liverpool SEM-EBSD system (using Oxford Instruments acquisition software) was tested using a kyanite single crystal (at installation, with Flamenco software) and more recently using a quartz single crystal (with AZtec software), both of known orientation. In all EBSD maps, the crystallographic orientation measured is consistent with the known orientation of the kyanite and quartz crystals in the SEM chamber (see also Kilian et al.[91]). Pole figures on the other hand display 'raw' orientations and no rotation to relate the SEM image to the EBSD data acquired is applied within AZtec and CHANNEL5 software. We therefore consistently apply a rotation of 180° to all pole figures produced using this software.

**Data availability.** The authors declare that all data supporting the findings of this study are available in the article and in Supplementary Information and Data files. Additional information is available from the corresponding author upon request.

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

## Acknowledgements

We acknowledge the support of the Starting Grant SLiM (306488) and Advanced Grant EVOKES (247076) of the European Research Council as well as the Deutsche Forschungsgemeinschaft grant LA2191/3-1. J.E.K. was funded by an Early Career Fellowship of the Leverhulme Trust (ECF 2016-325), E.M. and the EBSD-SEM laboratories at Liverpool Earth Sciences were supported by Natural Environment Research Council (NERC) grants NE/F018789/1, NE/L007363/1 and NE/M000060/1.

## Author contributions

J.E.K. conceptualised the study with Y.L. and D.B.D., and N.R.V. provided fieldwork assistance and expertise of Volcán de Colima. J.E.K. and Y.L. also performed the deformation experiments, J.E.K. conducted the EBSD data collection with E.M., processed the data with E.M. and J.W., and wrote the manuscript with the input of all authors.
