## [Peer Review File · Nature Communications]

Reviewers' comments:

Reviewer #1 (Remarks to the Author):

Thank you for the opportunity to review "Crystal plasticity as a strain marker of the viscous-brittle transition in magmas" by Kendrick et al.

The authors present an EBSD study of plagioclase microlites in as-sampled and experimentally deformed andesites from Volcán de Colima. The study tracks the evolution of crystal plasticity in the crystalline microlite fraction of the lavas with deformation, showing that crystal lattice misorientation increases with increasing stress and/or strain. The higher the crystal content and the lower the initial glass content, the more strain is accommodated by the crystals. Thus, the authors show evidence that, in crystal bearing magmas, strain is not only partitioned between the viscous interstitial melt and the eventual brittle failure of crystals, but that crystal plasticity also plays a role. The study concludes that crystal plastic deformation mechanisms cannot be discounted when considering how strain is accommodated in lavas and that crystal plasticity could be used as a strain marker.

The manuscript presents a gateway to a currently understudied aspect of magma rheology as it pertains to volcanology: the role of crystals in strain accommodation under conduit conditions. Generally, crystals are treated as exclusively rigid bodies in deforming magmas that may eventually break at high strain rates/stresses. The dataset presented here provides some tantalizing evidence that the role of crystals in the overall rheological behavior of magmas may be over-simplified. The study identifies a deformation mechanism that is largely ignored in eruption dynamics and may play an important role in the way volcanologists consider rheology.

From a technical point of view, the experiments are conducted under conditions appropriate for both magma ascent in the upper conduit and the crystal plastic deformation of plagioclase (though most deformation mechanism maps for plagioclase are constructed using experiments performed at very high pressures – this study does not use any confining pressure). The current manuscript is well written, concise, and generally self-consistent. There are some points that require a bit of clarification, but these are minor (please see my line comments below). The methods used are explicitly detailed and complete datasets are provided in the Supplementary Materials. The treatment of the previous literature is generally fair, though I have highlighted some minor omissions in the line comments below.

As acknowledged by the authors, further quantitative study of the importance of crystal plastic mechanisms in deforming magmas is needed to truly assess the significance of these mechanisms. However, I do not believe that this precludes publication of the current study in Nature Communications: indeed, I think it opens the door to an exciting avenue of study.

Below I have included 4 major comments that I believe would help improve the manuscript and a series of line comments that I hope the authors will find constructive and useful.

Major comments:

1) While I don't believe that the authors oversell their conclusion per se, I would caution against saying that the microlites act as 'strain markers' because they deform crystal plastically. Strain markers are used to aid in the quantification of strain, which is not what is done in this study. The microlites in this study are used to identify the activation of crystal plastic deformation mechanisms in the run up to brittle failure. How much strain is realistically accommodated by these crystals is not addressed. A simple rewording of 'strain markers' to 'indicators of strain' would make this clearer.

2) I think that a more rigorous discussion of the relative importance of crystal plastic deformation mechanisms in magmas can be conducted. For instance, Rutter et al., 2006 state that, though they observe dislocation movement, the strain accommodated by the crystal plastic deformation of quartz in a partially melted granitoid is negligible compared to the melt phase. They observe this in samples with initial glass contents generally lower than those in this study (<30%). I would have expected such a crystal plastic contribution to strain accommodation to increase with decreasing glass content (this appears to be confirmed by the present study) and, thus, the effect to be more pronounced in the Rutter et al., 2006 study than in the present study. I think it would be beneficial to the reader if the authors explicitly outlined why their conclusions differ from those of Rutter et al., 2006.

If, indeed, crystal plastic deformation mechanisms play an important role in the deformation of ascending magma, the natural follow-up question is: How much of the strain can be realistically accommodated by crystal plastic processes in the microlite phase of ascending magma? I would encourage the authors to address this in more detail, even if a subsequent study is ultimately needed.

3) The starting material has presumably been deformed during eruption. Thus, the microlites already have a relict crystallographic preferred orientation prior to being deformed experimentally. This does not necessarily detract from the study, as the authors are careful to characterize this (in Figures 1 and 2), but I think it would aid the reader if this was explicitly stated.

4) Figure 3 can be much better. The main (and enticing) conclusions of the study aren't particularly well represented by this final 'take-home' figure.

Firstly, the current figure does not present deformation mechanisms (as the caption states) but material behaviour. Further, I believe that the transition from viscous to crystal-plastic deformation is incorrect – it appears to describe the behavior of the magma in the centre of the conduit (viscous) but the behavior of the microlites near the conduit margins (crystal-plastic).

Secondly, I think the impact of the figure could be improved by proposing how the crystals themselves behave at various places in the conduit. It would be more instructive to focus on the strain partitioning between the microlites and the interstitial melt, then an overall 'volcano model'.

For instance: 1) the microlites likely behave rigidly in the centre of the conduit, where strain rates are low and strain is most likely accommodated by the interstitial melt; 2) the microlites start to deform crystal-plastically as one approaches the conduit margins, where strain rates/stresses increase and dislocation movement begins to be activated; and 3) the microlites are prone to breaking at the edge of the conduit margins, where strain rates and stresses are so high that the crystals need to break to continue accommodating strain. I think that a redrafted figure focusing on the crystals within the melt phase would do this study far more justice than the current Figure 3.

Minor comments:

Line 1: 'Strain marker' implies that you can quantify strain in some respect. Perhaps 'Crystal plasticity as an indicator of the viscous-brittle transition in magmas' is a more appropriate title.

Line 18: I find this statement a bit confusing. I take it that crystal plastic distortion is highest in the fragments of broken crystals, with respect to crystals that show no brittle deformation. Is this the case?

Line 21: "...bridging the viscous-brittle transition and leading crystal-bearing magmas towards failure."

Line 34: I think it would be appropriate to cite Arbaret et al., 2007 (JGR), Champallier et al., 2008 (EPSL), and Picard et al., 2013 (JGR) in your discussion of the rheology of crystal bearing magmas.

Line 39 and Line 40: I would suggest citing Laumonier et al., 2011 (Geology) as they demonstrate strain partitioning and shearing thinning in crystal bearing magmas. Furthermore, they demonstrate the development on Riedel shear structures in deforming crystal-bearing magmas.

Line 49: "Here, we demonstrate, for the first time that in magma, that strain..." This is simply a grammatical oversight. I would suggest: "Here we demonstrate, for the first time in magma, that strain..."

Line 51: I would clarify the sentence: "...between a network of rigid suspended particles that are susceptible to brittle failure."

Line 74: Is it necessary to specify 'silicic' if you also describe the glass as 'rhyolitic'?

Line 78: Are the cores oriented in the same direction? Could the authors please specify how the core orientations were chosen (with respect to foliation, perhaps?) and if the drilling orientations were the same for all samples.

Line 81 and Table 3: The strain rates quoted in the text do not appear consistent with the mean strain rates given in Table 3. Is this correct?

Line 87: 'brittle fracturing' is redundant. Please change to 'fracturing.'

Line 88: "The physical evolution of the samples included re-ordering of the porosity..." is the porosity truly being 're-ordered' if fractures are forming? Or is porosity being created?

Line 107: "These misorientations result from progressive rotations produced..." Rotations of what? Please specify.

Line 112: How was the 'original microlite orientation' chosen?

Line 114: Did the authors mean Figure 1E, instead of Figure 1D?

Lines 116-117: "The distortion within the fragments occurs along the same direction (Figure 1F)..." Is this true? The misorientation appears to increase with length in the two first fragments (starting from microlite length = 0), but decreases in the third, longest segment.

Line 121: "...all plagioclase microlites are elongated preferentially parallel to their a-axis..." What evidence is there of this and what is the reference frame used to judge this?

Line 155: Is the increase in lattice misorientation with increasing strain (from 20% to 30% strain; the red and blue lines in Figure 2) statistically significant?

Line 162: Does crystal plastic deformation continue after the crystals have broken?

Lines 162-163: "The reconstructed broken crystals record the highest misorientations observed in this study (Figure 2C), having more than twice the misorientation value..." Can the authors justify why the cumulative misorientation of the reconstructed crystals is appropriate? Does crystal plastic deformation not continue after the crystals are broken?

Line 169: 'brittle fracture' is redundant. Please replace with 'fracture'.

Line 216: "...magmas are visco-elastic fluids which are able to relax an applied stress..." Should this read "...are able to relax under an applied stress..."?

Lines 220 to 222: "...however extrusive volcanic products, such as lava domes (which form over longer timescales) must rely on other, as yet unidentified, indicators..." Lava domes don't need to rely on anything... Admittedly this is a picky comment, but perhaps it's worth rewording the sentence.

Line 225: Can you give an estimate of the crystal sizes (phenocrysts and microlites)? Were the EBSD step sizes appropriate for these crystal sizes? 15 microns seems large for typical microlite sizes...

Figure 1:

Please identify the principle stress axes in panel E. In panels C, G, D, and H: please specify the orientations of the pole axes with respect to the microstructure images (panels A and E).

I had some difficulty understanding what panels C and G were meant to show. Are they intended to highlight the changes in orientation of the principle slip directions from the natural material to the deformed material? If so, with how much confidence can the authors state the orientation of the slip systems in the starting material is consistent across all samples before deformation? Furthermore, the description of panel G states that it highlights the brittle fractures that have displaced the sample – is this by virtue of the data being grouped into several patches, as opposed to one (as in panel C)? Could this be explained a bit more thoroughly, please?

Line 492: "...with progressive rotation indicating crystal-plasticity resulting from dislocation..." Does the progressive rotation result from dislocation movement, creation? Please specify.

Line 493: Please label 'x' and 'y' on panel A (and also E).

Line 500: Do the authors mean to reference panel E here?

Line 502: Again, do the authors mean to reference panel E?

Figure 2:

Are the 'fragments' data shown in panel C the same data in A and B?

The legend in panel D is missing references to COLLB2 and COLLAH4.

Panel B: The change in misorientation of the lattices appears to be more affected by a smaller increase in stress (12MPa 30% for COLLAH4 compared to COLLB2). COLLAH4 contains more glass than COLLB2; I would have expected that such a response in the misorientation of the crystals would be more pronounced in a sample with less initial porosity and glass content. Can the authors comment on this?

In the article by Stunitz et al., (2003) (cited in the current study), those authors state that fracture almost always accompanies crystal plastic deformation in nature, with fracturing being a precursor to dislocation generation. If fracturing precedes dislocation generation, is it appropriate to reconstruct a total misorientation from crystal fragments? How is the misorientation of the entire crystal reconstructed?

Lines 522 and 523: "...higher misorientations in the denser COLB2..." What do the authors mean by 'denser'? Is it that COLB2 has both a lower initial porosity and a lower initial glass content, resulting in a more rigid magma overall? The mechanical data in Kendrick et al., (2013) appears to imply this. Perhaps the wording in the figure caption can be made more precise?

Table 3:

The mean strain rates reported in Table 3 are not the same as reported in the text. It appears that the data presented in the table are actually log(mean strain rate). The units should also be specified.

Should mean viscosity read 'apparent viscosity', to reflect the terminology in the text? Also, I believe the title should read $\log(\text{apparent viscosity})$.

Supplementary Materials:

Supplementary Figures S2 through S8: Please identify the principle stress axes on the figures (the principle compressive stress is only referred to in the caption of S3).

Figure S3:

A few questions/comments about the caption text:

"At the base, the more porous COLLAH4..." At the base of what? The SEM image or the figure?

"...applying stress/strain imparts fracturing." I think 'results in fracturing' would be more appropriate.

Please identify the fractures that have formed parallel to the compressive direction in the SEM images, perhaps with an arrow. They are much harder for the reader to identify in the COLLAH4 samples.

Figure S6: "(from the red cross at length 0/ the blue end)" – a typo?

Figure S8: Can the authors give a brief description of what the indexing colours mean? Does each indexed point represent the local crystal lattice orientation? If so, it's difficult to see the distortion with the crystals themselves – is this a matter of the colour scale being very large to encompass all indexed crystals? I expect if this was the case, then the misorientation of an individual crystal lattice would be drowned out – maybe this is worth stating.

Figure S9: "...suggesting that microlites become more elongate as they grow..." Are the microlites growing during deformation? Is this mentioned in the main text?

Figure S10: "This verifies a more systematic..." I would contend that it "suggests a more systematic...", not verifies.

Kind regards,
Alexandra Kushnir

Reviewer #2 (Remarks to the Author):

Review of "Crystal plasticity as a strain marker of the viscous-brittle transition in magmas", by Kendrick et al.

In this work, the authors analysed the lattice misorientation of plagioclase microlite in naturally and experimentally deformed andesite magmas. Based on the increase in the misorientation with stress, the authors infer that crystal plasticity can be used as a deformation marker.

This work is a valuable contribution to the understanding of the lava effusion mechanism, as observed at Mt. St. Helens and Mt. Unzen. This quantitative description of the misorientation in the microlite in naturally and experimentally deformed lava is the first attempt of its kind, and should inspire future work. However, there are some important issues that should be addressed prior to publication. Without resolving these issues, this paper should not be accepted for publication in Nature Communications.

General comments

1. This paper reports a quantitative description of the lattice misorientation found in microlite plagioclase. However, the importance of this finding is unclear. The authors first propose that plastic deformation controls magma rheology (L 203–205). However, the role of plastic deformation of tiny crystals, i.e., microlite, on magma rheology and lava effusion is unclear. I do not think that plastic deformation controls magma flow in a volcanic conduit because the shear-localized and fractured weak zone determines flow in the conduit (Tuffen and Dingwell, 2005; Cashman et al., 2008). Second, the transition process from viscous to crystal-plastic to brittle deformation is unclear (Fig. 3). Under high strain rate, silicate melt shows solid-like behaviour (Dingwell, 1996). In contrast, magma with high crystallinity indicates solid-like behaviour due to crystal interaction. The model presented in Fig. 3 includes both the processes. The authors need to explain the details of the transition processes. Finally, the authors propose that the lattice misorientation can be used as a strain maker, but the data show that the misorientation is independent of strain (Fig. 2). Therefore, the misorientation cannot be used as the strain maker. As a result, the importance of this study is unclear, and the authors need to clarify the implications and significance of their result.

2. The two interpretations presented for the analytical data are difficult to understand. First, the data in Fig. 2 show large scattering. The authors simply used a linear fitting on the data. Using this fitting, they obtained an important parameter, the average misorientation per length; however, they completely neglected discussing the error in this parameter. As pointed out previously, the data show a large scatter; hence, the simple linear fit cannot be accepted without a quantification of error. The interpretation of brittle to plastic deformation transition of plagioclase microlite is also difficult to understand. The authors inferred that the plastic deformation results in brittle fracturing at a threshold for maximum lattice misorientation. However, the fractured microlite in Fig. 1E (central fragment) does not show misorientation. In addition, in the Supporting Information (Figures S6–S8), many fractured crystals do not show misorientation. Without additional explanation, it appears that the authors are over-interpreting their data.

3. The relationship between deformation and misorientation is unclear because the experiments were performed using naturally deformed andesite. In a large undercooling system, lattice misorientation in clinopyroxene forms in the crystallization process, not deformation (e.g., Hammer et al., 2010). Brugger and Hammer (2015) reported that no misorientation was found in plagioclase that was formed in crystallization experiments. Based on the previous data, the authors need to discuss the formation of misorientation during crystallization. In addition, I strongly suggest that the authors compare the analytical results from natural samples with different degrees of deformation, i.e., non-deformed lava and shear-zone lava. All of these data support the interpretations presented in this study.

4. It would be helpful to provide the details of the EBSD experimental design. The absolute orientations from EBSD measurements depend on the design and implementation of the experiment (e.g., Kilian et al., 2016). This is just a confirmation; however, the data reliability will be supported by the explanation.

Other comments

Line 73: Please provide the chemical composition of the plagioclase phenocryst and microlite.

Line 76: How did you determine the error in the pores? On the other hand, why do the solid portions have no error?

Line 88: Have the authors measured the porosity and permeability? This data would also be interesting.

Line 143: To identify the trend, an estimation of the error is necessary (see general comment 2).

Line 154: Figure 2 clearly indicates that the strain does not cause an increase in the misorientation. Why do the authors consider the misorientation as a strain maker? (see general comment 1).

Line 169: Quantitative data and error estimation need to be provided to indicate the size reduction, i.e., the crystal size distribution should be measured and provided.

Line 198: Brittle fracturing does not seem to involve plastic deformation (see general comment 2).

Line 214: 'stress/strain' is incorrect, because stress controls the formation of the misorientation while the misorientation is independent of strain. Stress and strain are different, and this expression is very confusing.

Line 224: Misorientation cannot be used as a strain maker, although it depends on stress.

Line 234: How were the contents measured? How were the images obtained? SEM? Optical microscope?

Line 237: Please verify the crystallization during heating and deformation. The analyses for samples which were heated but not deformed are also necessary. Ideally, this data should be compared with data obtained from the deformation experiment, because the natural sample does not include the effect of heating.

Line 252: Tilt is 70°?

Supplementary Figures 7 & 8: Plagioclase microlite shows twinning. The formation mechanism of the twinning should be discussed.

References

Brugger, C.R. and J.E. Hammer (2015) Prevalence of growth twins among anhedral plagioclase microlite. *American Mineralogist*, 100, 385–395.

Cashman, K.V., C.R. Thornber and J.S. Pallister (2008) From dome to dust: Shallow crystallization and fragmentation of conduit magma during the 2004 – 2006 dome extrusion of Mount St. Helens, Washington.

Dingwell, D.B. (1996) Volcanic dilemma: Flow or blow? *Science*, 273, 1054–1055.

Hammer, J.E., T.G. Sharp and P. Wessel (2010) Heterogeneous nucleation and epitaxial crystal growth of magmatic minerals. *Geology*, 38, 367–370.

Kilian, R., M. Bestmann and R. Heilbronner (2016) Absolute orientations from EBSD measurements - as easy as it seems? *Geophysical Research Abstracts*, 18, EGU2016-8221.

Tuffen, H. and D. Dingwell (2005) Fault textures in volcanic conduits: evidence for seismic trigger mechanism during silicic eruptions. *Bulletin Volcanology*, 67, 370–387.

Below we provide our replies in blue to all the reviewer comments.

Reviewers' comments:

Reviewer #1 (Remarks to the Author):

Thank you for the opportunity to review “Crystal plasticity as a strain marker of the viscous-brittle transition in magmas” by Kendrick et al.

The authors present an EBSD study of plagioclase microlites in as-sampled and experimentally deformed andesites from Volcán de Colima. The study tracks the evolution of crystal plasticity in the crystalline microlite fraction of the lavas with deformation, showing that crystal lattice misorientation increases with increasing stress and/or strain. The higher the crystal content and the lower the initial glass content, the more strain is accommodated by the crystals. Thus, the authors show evidence that, in crystal bearing magmas, strain is not only partitioned between the viscous interstitial melt and the eventual brittle failure of crystals, but that crystal plasticity also plays a role. The study concludes that crystal plastic deformation mechanisms cannot be discounted when considering how strain is accommodated in lavas and that crystal plasticity could be used as a strain marker.

The manuscript presents a gateway to a currently understudied aspect of magma rheology as it pertains to volcanology: the role of crystals in strain accommodation under conduit conditions. Generally, crystals are treated as exclusively rigid bodies in deforming magmas that may eventually break at high strain rates/stresses. The dataset presented here provides some tantalizing evidence that the role of crystals in the overall rheological behavior of magmas may be over-simplified. The study identifies a deformation mechanism that is largely ignored in eruption dynamics and may play an important role in the way volcanologists consider rheology.

From a technical point of view, the experiments are conducted under conditions appropriate for both magma ascent in the upper conduit and the crystal plastic deformation of plagioclase (though most deformation mechanism maps for plagioclase are constructed using experiments performed at very high pressures – this study does not use any confining pressure). The current manuscript is well written, concise, and generally self-consistent. There are some points that require a bit of clarification, but these are minor (please see my line comments below). The methods used are explicitly detailed and complete datasets are provided in the Supplementary Materials. The treatment of the previous literature is generally fair, though I have highlighted some minor omissions in the line comments below.

As acknowledged by the authors, further quantitative study of the importance of crystal plastic mechanisms in deforming magmas is needed to truly assess the significance of these mechanisms. However, I do not believe that this precludes publication of the current study in Nature Communications: indeed, I think it opens the door to an exciting avenue of study.

Below I have included 4 major comments that I believe would help improve the manuscript and a series of line comments that I hope the authors will find constructive and useful.

We thank the reviewer, Dr Alexandra Kushnir, for her concise summary and for her carefully considered comments, which we have taken time to address comprehensively.

Major comments:

1) While I don't believe that the authors oversell their conclusion per se, I would caution against saying that the microlites act as 'strain markers' because they deform crystal plastically. Strain markers are used to aid in the quantification of strain, which is not what is done in this study. The microlites in this study are used to identify the activation of crystal plastic deformation mechanisms in the run up to brittle failure. How much strain is realistically accommodated by these crystals is not addressed. A simple rewording of 'strain markers' to 'indicators of strain' would make this clearer. The authors agree that we do not quantify the amount of strain accommodated by the microlites, indeed this was not the goal of the study – rather, we use the amount of crystal plasticity to indicate the magnitude of strain experienced by the magma (since melt itself is infinitely deformable and hence does not record the strain history). This was our reasoning behind the term “strain marker” since it evidences a record of strain, but the authors realise that this phrasing was misleading in this context, and have changed the phrase to “indicators of strain” throughout. We make 2 exceptions to this, at the end of the abstract and in the discussion, where we discuss the possibility that, with future work and quantification crystals might serve as strain markers within certain well-constrained scenarios. We hope that the description of our work, and the goals of the study are now clearer thanks to these changes.

2) I think that a more rigorous discussion of the relative importance of crystal plastic deformation mechanisms in magmas can be conducted. For instance, Rutter et al., 2006 state that, though they observe dislocation movement, the strain accommodated by the crystal plastic deformation of quartz in a partially melted granitoid is negligible compared to the melt phase. They observe this in samples with initial glass contents generally lower than those in this study (<30%). I would have expected such a crystal plastic contribution to strain accommodation to increase with decreasing glass content (this appears to be confirmed by the present study) and, thus, the effect to be more pronounced in the Rutter et al., 2006 study than in the present study. I think it would be beneficial to the reader if the authors explicitly outlined why their conclusions differ from those of Rutter et al., 2006.

We have added a discussion of the Rutter et al observations at lines and To briefly summarise, we firstly would like to emphasise that the Rutter et al. (2006) study has different aims than ours, focussing on measuring experimentally the deformation mechanisms in partially molten granites, where lithologies are dominated by a granitic melt and quartz clasts. These authors measure a stress exponent n between 1 and 2 and interpret it as attributable to the viscous behaviour of the weak melt, with some additional, not fully understood behaviour that causes deviations from fully linear viscosity. Mecklenburgh et al. (2003), studying a similar system, do show some evidence for crystal plasticity in the quartz grains, which could be the cause of $n > 1$, but conclude that this has little influence on the rock rheology. We do not focus on whole rock rheology, but rather observe the behaviour of microlites, whereby the window of conditions within which they deform plastically may be a good indicator of the viscous-to-brittle transition in magmas.

If, indeed, crystal plastic deformation mechanisms play an important role in the deformation of ascending magma, the natural follow-up question is: How much of the strain can be realistically accommodated by crystal plastic processes in the microlite phase of ascending magma? I would encourage the authors to address this in more detail, even if a subsequent study is ultimately needed.

It's important to note a few things here, which presumably were not made completely clear in the

manuscript (we have endeavoured to clarify this in the manuscript). First, we do not believe that the strain accommodated by the crystals themselves is a particularly high fraction of the strain accommodated by deforming magma – however, we do believe that the deformation recorded in the microlites is proportional to the deformation experienced by the bulk magma – and this is key – crystal plasticity can help identify areas of (and proportions of magma affected by) strain localisation, for example in dome eruptions (where the bulk may extrude relatively undeformed, mantled by shear damage zones). Moreover, a second feature of crystal plasticity is with regard to maximum packing fraction, which influences a magma’s propensity to “flow or blow” - crystals yielding allow magmas to avoid “locking” of the rigid crystal framework and hence might favour flowing more if crystals can bend/ deform around one another, facilitating alignment which would lower viscosity – this could also be important in other settings, for example aiding mushes to mobilize, and importantly when they yield, grain size distribution evolves and max packing changes, further changing viscosity. We have edited the discussion of the importance of this.

3) The starting material has presumably been deformed during eruption. Thus, the microlites already have a relict crystallographic preferred orientation prior to being deformed experimentally. This does not necessarily detract from the study, as the authors are careful to characterize this (in Figures 1 and 2), but I think it would aid the reader if this was explicitly stated.

The previous version of the manuscript made no reference to crystallographic preferred orientation (CPO), but stated that initial material deformed during its eruptive history, and as such, it contained evidence for crystal plasticity. In the previous figures 1 and 2 (now 4 and 5) what we’re showing is the lattice distortion of the microlites (which is an intra-crystalline deformation), and rather we do not discuss CPO (which is a bulk distortion / flow indicator). Our reasoning being that the scale of the areas analysed was rather small in comparison to features of the magma (phenocrysts and pores), which are likely to cause local perturbations in CPO. Further, rocks were all cored in the same direction within each block so each sample suite is comparable even if there was pre-existing CPO at a scale larger than that which we could detect. That said, with the reviewer’s comment in mind, and the point from the editor that we could add further supporting figures, we have performed some new analysis and created a new figure exploring the CPO of the microlites. The new data (now shown in fig. 3, shown right) demonstrates the local alignment of microlites, particularly around phenocrysts, and particularly in the higher crystallinity COLB2 sample – this confirms that at the scale examined we cannot distinguish a bulk CPO common across samples, and may help explain the variability in plasticity experienced within one sample due to the local orientation with respect to the principal stress direction (indicated in the new fig. 6).

4) Figure 3 can be much better. The main (and enticing) conclusions of the study aren't particularly well represented by this final 'take-home' figure.

Firstly, the current figure does not present deformation mechanisms (as the caption states) but material behaviour. Further, I believe that the transition from viscous to crystal-plastic deformation is incorrect – it appears to describe the behavior of the magma in the centre of the conduit (viscous) but the behavior of the microlites near the conduit margins (crystal-plastic).

Secondly, I think the impact of the figure could be improved by proposing how the crystals themselves behave at various places in the conduit. It would be more instructive to focus on the strain partitioning between the microlites and the interstitial melt, then an overall 'volcano model'. For instance: 1) the microlites likely behave rigidly in the centre of the conduit, where strain rates are low and strain is most likely accommodated by the interstitial melt; 2) the microlites start to deform crystal-plastically as one approaches the conduit margins, where strain rates/stresses increase and dislocation movement begins to be activated; and 3) the microlites are prone to breaking at the edge of the conduit margins, where strain rates and stresses are so high that the crystals need to break to continue accommodating strain. I think that a redrafted figure focusing on the crystals within the melt phase would do this study far more justice than the current Figure 3.

We agree with the reviewer, we have changed the conduit schematic, added inserts / expansions of the micro-scale behaviour at each point, and more detailed annotation (now figure 7). The figure now appears as follows:

Minor comments:

Line 1: 'Strain marker' implies that you can quantify strain in some respect. Perhaps 'Crystal plasticity as an indicator of the viscous-brittle transition in magmas' is a more appropriate title.

We agree that this title is representative of what we aim to present and have changed it accordingly.

Line 18: I find this statement a bit confusing. I take it that crystal plastic distortion is highest in the fragments of broken crystals, with respect to crystals that show no brittle deformation. Is this the case?

Yes, the highest distortions in fragments of subsequently broken bits. We have slightly edited the abstract in light of other comments too and hope this is clearer.

Line 21: "...bridging the viscous-brittle transition and leading crystal-bearing magmas towards failure."

We have slightly edited the abstract.

Line 34: I think it would be appropriate to cite Arbaret et al., 2007 (JGR), Champallier et al., 2008 (EPSL), and Picard et al., 2013 (JGR) in your discussion of the rheology of crystal bearing magmas. We apologise for the omission, through many iterations these were left out and have been added at the relative points for discussion.

Line 39 and Line 40: I would suggest citing Laumonier et al., 2011 (Geology) as they demonstrate strain partitioning and shearing thinning in crystal bearing magmas. Furthermore, they demonstrate the development on Riedel shear structures in deforming crystal-bearing magmas.

Agreed, we have cited this very relevant work here and elsewhere.

Line 49: "Here, we demonstrate, for the first time that in magma, that strain..." This is simply a grammatical oversight. I would suggest: "Here we demonstrate, for the first time in magma, that strain..."

This has been corrected.

Line 51: I would clarify the sentence: "...between a network of rigid suspended particles that are susceptible to brittle failure."

This has been corrected.

Line 74: Is it necessary to specify 'silicic' if you also describe the glass as 'rhyolitic'?

Agreed, it should be either-or!

Line 78: Are the cores oriented in the same direction? Could the authors please specify how the core orientations were chosen (with respect to foliation, perhaps?) and if the drilling orientations were the same for all samples.

Yes, the cores were all cut from the blocks in the same direction, right next to one another. The thin sections were then cut parallel to principal stress direction, and also the same direction was used for the thin sections of the starting materials. The blocks, the cores cut from them and the thin sections showed no visible signs of magmatic flow textures (e.g., flow banding). As detailed above, we have now examined the dataset anew to look for foliation, or CPO in the microlites, which we find is affected by local perturbations due to phenocrysts, but is not noticeably systematic across the sample suite.

Line 81 and Table 3: The strain rates quoted in the text do not appear consistent with the mean strain rates given in Table 3. Is this correct?

Apologies, we made a mistake on the unit in the table which made the strain rates look much faster – this has been corrected and the values are the same as in the text (to less significant figures for ease of reading).

Line 87: 'brittle fracturing' is redundant. Please change to 'fracturing.'

We agree that this may be redundant for experts/ experienced researchers in the relevant field, but we do not believe it detracts from the meaning, and believe it will aid visualisation for non-experimentalists who may otherwise envisage more of a flow-like tearing, rather than a sharp,

distinct fracture. A quick survey of the literature suggests this is a commonly employed phrasing.

Line 88: “The physical evolution of the samples included re-ordering of the porosity...” is the porosity truly being ‘re-ordered’ if fractures are forming? Or is porosity being created?

We see the reviewer’s point, if total porosity remains the same value but is in different arrangement then can be argued as re-ordering, however, we have reworded this to make the observation clearer.

Line 107: “These misorientations result from progressive rotations produced...” Rotations of what? Please specify.

This meant rotations of the crystal lattice, we’ve actually produced a new figure which helps visualise this process (now figure 1 shown right) and we’ve rephrased for clarity. The new figure shows the development of a dislocation, and shows the dislocation passing through the sample.

Line 112: How was the ‘original microlite orientation’ chosen?

This text was misleading, what was meant was that we chose one EBSD data pixel at the end of a microlite as the reference point (the absolute orientation is not considered) and we plot the intracrystalline (i.e. lattice) distortion from that reference point - we’re highlighting distortion, or the amount of bending of the microlite, without considering where it sits relative to anything else. We have rephrased the sentence accordingly.

Line 114: Did the authors mean Figure 1E, instead of Figure 1D?

Yes, thanks for pointing this out, in fact the following couple of figure references pointed to the wrong panels due to late edits, as you will see in the next comment.

Lines 116-117: “The distortion within the fragments occurs along the same direction (Figure 1F)...” Is this true? The misorientation appears to increase with length in the two first fragments (starting from microlite length = 0), but decreases in the third, longest segment.

This was unclear – first, we attempted to refer to Figure 1G –the pole figure demonstrates the axes along which the distortion took place, i.e. by looking at the similarity in the spread of each fragment of the crystal on each panel of the pole figure you see this. This is verified by Figure 1H, which shows that the slip systems of the crystals for the areas marked in 1E, and you can see that area 4 (on a short fragment with “increasing misorientation with length”) and area 1 (on the long fragment with “decreasing misorientation with length”) have the same result (centred on [001]). Incidentally, in 1F

Figure 1: Schematic representation of a dislocation in a crystal lattice. Here an edge dislocation forms and propagates through the lattice for simplicity, but screw dislocations are also common. The stages show: (A) Isostatic stress conditions, and no deformation; (B) Differential stress applied, leading to elastic strain; (C) Yielding occurs under applied stress, creating a dislocation; (D) The dislocation migrates under differential stress conditions; (E) The dislocation passes through, resulting in distortion and (F) Upon return to isostatic stress conditions, elastic strain is recovered. Dislocations that have passed through as in F leave a shape change but no internal distortion, however, significant densities of dislocations stranded within the lattice at stages C-D give rise to macroscopic lattice distortion as documented here.

(the graph) the change in slope (two positive, one negative as the reviewer puts it) is likely due to a more complex shear geometry – we rephrased the section which refers to this figure.

Line 121: “...all plagioclase microlites are elongated preferentially parallel to their a-axis...” What evidence is there of this and what is the reference frame used to judge this?

This means with respect to the a-b-c axis of the crystal system, i.e. the basis vectors of the lattice. In plagioclase, growth (or the long axis) tends to occur predominantly along c, however, magmatic plagioclase microlite is typically elongate along a – this has been documented (see e.g. Deer et al 2013) although the reasoning is unknown. We see this in our samples plotting the axes direction from the EBSD data, which records the 3D crystallographic information of a mineral, despite being only on a 2D surface. We have now added this as a figure (supplementary fig. S10), showing the crystallographic axes when a microlite is dissected along its long axis or short axis (pasted here). Note that in sub-equant grains (i.e. elongated orthogonal to the plane of the page) the a axis is orthogonal to the plane of the page (Fig. S10 B). On the other hand, in elongated grains, the a axis is parallel to the elongation direction (Fig. S10A).

Line 155: Is the increase in lattice misorientation with increasing strain (from 20% to 30% strain; the red and blue lines in Figure 2) statistically significant?

Based on a couple of queries from both reviewers we decided to perform some further statistical analysis of our dataset. Using the metric “misorientation per micron” which is simply the lattice distortion (in degrees) divided by the crystal length (in microns) as documented in the manuscript, we investigated the maximum, minimum, median, and plotted these on a box-plot (new figure 6) along with the 25th and 75th percentiles of the values. Using the misorientation per micron allows us to exclude any influence of measuring larger or smaller crystals in any sample.

The box-plot shown to the right gives the results of the further analysis and demonstrates a number of important points. With regard to the above question, we can see that the mean value (x on the plots) of misorientation per micron increases as deformation conditions increase (from left to right) in both samples, as does the minimum value and the maximum value – so, increasing strain at the same stress does increase all 3 of these in a statistically relevant manner. Interestingly though, in both the samples the median, and both the 25th and 75th percentiles change very little from 20 to 30% strain at 28MPa, suggesting that while the deformation of some microlites certainly increases (the max, min and mean increase), some of the microlites may not – this could be due to strain partitioning coming into effect after a certain amount of strain, such that not all microlites are still suffering deformation. This would be a really interesting avenue to explore, perhaps with in-situ synchrotron experiments, or with higher time-resolution investigations (and perhaps a simpler system).

Line 162: Does crystal plastic deformation continue after the crystals have broken?

One has to assume yes, but this will depend on the stress-field, which would presumably need to develop anew if the framework was altered by the microlite breaking (i.e. the fracturing relieved the stress that was building up, and allowed the crystal to move freely again). Given this and a few other comments, we no longer “reconstruct” the broken crystals, we just consider the fragments (see next answer).

Lines 162-163: “The reconstructed broken crystals record the highest misorientations observed in this study (Figure 2C), having more than twice the misorientation value...” Can the authors justify why the cumulative misorientation of the reconstructed crystals is appropriate? Does crystal plastic deformation not continue after the crystals are broken?

We have taken this point on board and considered it carefully. By performing the new statistical analysis shown above (in the boxplot, figure 6 in the manuscript) we hope that we demonstrate more effectively how significant the difference in the deformation is in the broken crystals. With this in mind we have edited figure 2 (now fig. 5) which summarises the deformation in each sample set, including the broken fragments, which we now incorporated into panels A and B of figure 2 (now fig. 5). We no longer sum the fragments into reconstructed crystals, and in fact this served to reduce the

scatter and increase the gradient of the misorientation per micron slope shown in panel D (now C) of figure 2 (now 5). Along with the new figure 6 we believe we demonstrate clearly that the broken crystals suffered by far the most deformation, and we conclude that this points to a mechanism where plastic deformation accrued, leading toward brittle failure after a limit was exceeded.

Line 169: 'brittle fracture' is redundant. Please replace with 'fracture'.

See previous reply regarding this.

Line 216: "...magmas are visco-elastic fluids which are able to relax an applied stress..." Should this read "...are able to relax under an applied stress..."?

No, "relax an applied stress" is accurate here, it is also true that they can relax under an applied stress, but here we refer to dissipating a given applied stress so use the former.

Lines 220 to 222: "...however to unravel the history of extrusive volcanic products, such as lava domes (which form over longer timescales) must rely on other, as yet unidentified, indicators..." Lava domes don't need to rely on anything... Admittedly this is a picky comment, but perhaps it's worth rewording the sentence.

Changed as suggested.

Line 225: Can you give an estimate of the crystal sizes (phenocrysts and microlites)? Were the EBSD step sizes appropriate for these crystal sizes? 15 microns seems large for typical microlite sizes... The microlite sizes are best shown by Figure 3, the new supplementary Figure S7 – there is no need to estimate these, all the data is shown in the supplementary tables S2-9 - they range from 0-70microns, and were measured with a step size of 0.2 microns (not 15 microns), which is very high spatial resolution for EBSD on geological materials. The phenocrysts however are much larger, as shown in Figure 2 (new) and Supplementary Figures S2, S4 and S5 – 0.2 to 2mm in length, and these were imaged at much coarser resolution (15 microns) to allow to map a larger area and avoid unnecessary accumulation of data – we deem this was ample to demonstrate the fracture of the phenocrysts in fig s6, with transects still comprised of some 200 points.

Figure 1:

Please identify the principle stress axes in panel E.

Done.

In panels C, G, D, and H: please specify the orientations of the pole axes with respect to the microstructure images (panels A and E).

This has been added in the caption.

I had some difficulty understanding what panels C and G were meant to show. Are they intended to highlight the changes in orientation of the principle slip directions from the natural material to the deformed material? If so, with how much confidence can the authors state the orientation of the slip systems in the starting material is consistent across all samples before deformation?

No, this is not what they show. This is in-part why we added the new figure 3 now (pasted above in a previous reply) showing the analysed areas (in band contrast) along with pole figures for all microlites. These (new) pole figures in fact show the orientations of all microlites across all samples, in the sample reference frame, highlighting local perturbations in orientation resulting from heterogeneities such as phenocrysts and pores. Panels C and G rather show one such microlite on each, colour coded according to the texture component (as described in the manuscript) in panels A

and E, showing their intra-crystalline (internal to the grain) distortion, their absolute orientation within the sample is not discussed.

Note that slip directions are part of a slip system (composed by a slip plane and a slip direction) and specific slip systems are characteristic of specific materials and are activated by the level of stress (which is likely to be local in this study) seen by a given microlite (or a group of microlites, as they appear to cluster in these andesites). Slip system activity is also a function of the orientation of a microlite with respect to the stress seen. Here we do not attempt to discuss slip systems across samples, nor do we discuss crystallographic preferred orientations (CPOs) as these are a different aspect to explore, interesting but not the focus of this study. As a further comment we would like to add that in the systems studied here CPOs would not be the result (or linked to) crystal plasticity, but would mainly be the consequence of passive rotation of microlites due to magma flow.

Furthermore, the description of panel G states that it highlights the brittle fractures that have displaced the sample – is this by virtue of the data being grouped into several patches, as opposed to one (as in panel C)? Could this be explained a bit more thoroughly, please?

Yes each patch on C and G corresponds to the fragment of the same colouring in panels A and E respectively. A gradual colour change represents a small change in misorientation, while a sharp change in colour represents a sharper change in misorientation, as across the fracture in panel E. This has been expressed clearly in the caption now.

Line 492: “...with progressive rotation indicating crystal-plasticity resulting from dislocation...” Does the progressive rotation result from dislocation movement, creation? Please specify.

To say the movement was progressive implied a time constrain which we could not observe, hence we’ve removed this.

Line 493: Please label ‘x’ and ‘y’ on panel A (and also E).

We tried to remove the x-y as it was confusing, but missed it in the caption. We have now better explained where the profile was measured, as seen in panels A and E.

Line 500: Do the authors mean to reference panel E here?

Yes, we apologise for the mistake.

Line 502: Again, do the authors mean to reference panel E?

Yes, we apologise for the mistake.

Figure 2:

Are the ‘fragments’ data shown in panel C the same data in A and B?

No these are not in the other datasets. In the main analysis broken crystals are only included if only one fragment was substantial and could be indexed while the other fragments could not – e.g. crystal no. 31 in Table S3.

The legend in panel D is missing references to COLLB2 and COLLAH4.

This figure and caption are now different (see other comments regarding broken crystals), and the key is now complete – the data is now shown in Figure 5.

Panel B: The change in misorientation of the lattices appears to be more affected by a smaller increase in stress (12MPa 30% for COLLAH4 compared to COLLB2). COLLAH4 contains more glass

than COLB2; I would have expected that such a response in the misorientation of the crystals would be more pronounced in a sample with less initial porosity and glass content. Can the authors comment on this?

We agree with the comment, intuition suggests the lower porosity, lower glass content sample would be more susceptible. We note that the absolute deformation values are still lower in COLLAH4 than COLB2 (after 30% strain at 16 MPa), so could be that the natural LAH4 was under-deformed with respect to B2 (LAH4 certainly has lower initial deformation), rather than being more influenced by stress than B2.

In the article by Stunitz et al., (2003) (cited in the current study), those authors state that fracture almost always accompanies crystal plastic deformation in nature, with fracturing being a precursor to dislocation generation. If fracturing precedes dislocation generation, is it appropriate to reconstruct a total misorientation from crystal fragments? How is the misorientation of the entire crystal reconstructed?

See the previous answer regarding this query too; we had used the reconstructed crystals as a statistical point to try to find where the yield point is, i.e. max misorientation possible, rather than expressing total pre-fracture misorientation. However, as we said before we've now removed this and hope that we made this point more clearly with the new box-plot.

Lines 522 and 523: "...higher misorientations in the denser COLB2..." What do the authors mean by 'denser'? Is it that COLB2 has both a lower initial porosity and a lower initial glass content, resulting in a more rigid magma overall? The mechanical data in Kendrick et al., (2013) appears to imply this. Perhaps the wording in the figure caption can be made more precise?

We simply meant lower porosity, and have reworded accordingly.

Table 3:

The mean strain rates reported in Table 3 are not the same as reported in the text. It appears that the data presented in the table are actually $\log(\text{mean strain rate})$. The units should also be specified. Should mean viscosity read 'apparent viscosity', to reflect the terminology in the text? Also, I believe the title should read $\log(\text{apparent viscosity})$.

The seemingly different strain rate was due to an error in the units of the strain rate column, which has now been corrected.

Supplementary Materials:

Supplementary Figures S2 through S8: Please identify the principle stress axes on the figures (the principle compressive stress is only referred to in the caption of S3).

Done.

Figure S3:

A few questions/comments about the caption text:

“At the base, the more porous COLLAH4...” At the base of what? The SEM image or the figure?

We just meant at the base of the figure, we now added panel A and B to clarify.

“...applying stress/strain imparts fracturing.” I think ‘results in fracturing’ would be more appropriate.

Agreed

Please identify the fractures that have formed parallel to the compressive direction in the SEM images, perhaps with an arrow. They are much harder for the reader to identify in the COLLAH4 samples.

We tried a few ways of highlighting the fractures, but all of them obscured the fractures themselves or left the image looking cluttered, as such we have not added any annotation, but have added more detail in the caption.

Figure S6: “(from the red cross at length 0/ the blue end)” – a typo?

Correct but unclear, we have reworded.

Figure S8: Can the authors give a brief description of what the indexing colours mean? Does each indexed point represent the local crystal lattice orientation? If so, it’s difficult to see the distortion with the crystals themselves – is this a matter of the colour scale being very large to encompass all indexed crystals? I expect if this was the case, then the misorientation of an individual crystal lattice would be drowned out – maybe this is worth stating.

Yes, these All-Euler angles use RGB colour coding, whereby the combination of 3 Euler angles that takes us from the SEM reference frame to the orientation of crystal, is visualised as one final combined colour (e.g. purple). Subtle misorientations (<5 degrees) within that crystal cannot be seen clearly using the All-Euler colour scheme (because large changes in absolute orientations are being measured). This is where the texture component too becomes useful (previously in Figure 1, now fig 4) as it allows us to investigate intracrystalline distortions of individual crystals where colour coding can be set to range from blue to red across small misorientations, using a reference point within the same crystal. We added a comment about the all-Euler colours in the methods.

Figure S9: “...suggesting that microlites become more elongate as they grow...” Are the microlites growing during deformation? Is this mentioned in the main text?

No they are not growing, just that longer (bigger) crystals also have higher aspect ratios in all samples i.e. as they grew they became more elongate. – we have tried to express this more clearly in the caption.

Figure S10: “This verifies a more systematic...” I would contend that it “suggests a more systematic...”, not verifies.

Changed.

Kind regards,
Alexandra Kushnir

Reviewer #2 (Remarks to the Author):

Review of “Crystal plasticity as a strain marker of the viscous-brittle transition in magmas”, by Kendrick et al.

In this work, the authors analysed the lattice misorientation of plagioclase microlite in naturally and experimentally deformed andesite magmas. Based on the increase in the misorientation with stress, the authors infer that crystal plasticity can be used as a deformation marker.

This work is a valuable contribution to the understanding of the lava effusion mechanism, as observed at Mt. St. Helens and Mt. Unzen. This quantitative description of the misorientation in the microlite in naturally and experimentally deformed lava is the first attempt of its kind, and should inspire future work. However, there are some important issues that should be addressed prior to publication. Without resolving these issues, this paper should not be accepted for publication in Nature Communications.

General comments

1. This paper reports a quantitative description of the lattice misorientation found in microlite plagioclase. However, the importance of this finding is unclear. The authors first propose that plastic deformation controls magma rheology (L 203–205). However, the role of plastic deformation of tiny crystals, i.e., microlite, on magma rheology and lava effusion is unclear. I do not think that plastic deformation controls magma flow in a volcanic conduit because the shear-localized and fractured weak zone determines flow in the conduit (Tuffen and Dingwell, 2005; Cashman et al., 2008). Shear localized and fracture zones are exactly where this is important, the plasticity marks the transition to this strain localised zone (which is not as straight-forward in suspensions as it is in obsidian i.e. Tuffen and Dingwell, 2005) – we are proposing, not that the microlites hold significant proportions of the strain themselves, nor that crystal plasticity regulates the viscosity to a first order, but that they indicate where the strain has been localised in the suspension by deforming plastically, and they indicate approximately the magnitude of the strain suffered in that region by becoming increasingly plastically deformed as stress/strain increase. The melt itself is unable to record this information unless it is quenched rapidly, as at longer timescales the melt will simply relax and recover from any applied deformation.

Second, the transition process from viscous to crystal-plastic to brittle deformation is unclear (Fig. 3). Under high strain rate, silicate melt shows solid-like behaviour (Dingwell, 1996). In contrast, magma with high crystallinity indicates solid-like behaviour due to crystal interaction. The model presented in Fig. 3 includes both the processes. The authors need to explain the details of the transition processes.

We agree that the figure did not entirely depict what we hoped to show, (although the presence of crystals in magmatic suspensions decrease the strain rate limits to achieve failure, as defined for liquids in Dingwell, 1996 (also see Cordonnier et al., 2012)) in short, that crystal-plasticity is a necessary stage preceding failure. This was largely because the schematic figure failed to differentiate between crystal behaviours and bulk suspension behaviours. We have edited this figure (now figure 7), which now expresses our findings more clearly (please see other reply to reviewer 2 later in the responses for details of the edits).

Finally, the authors propose that the lattice misorientation can be used as a strain marker, but the data show that the misorientation is independent of strain (Fig. 2). Therefore, the misorientation

cannot be used as the strain maker. As a result, the importance of this study is unclear, and the authors need to clarify the implications and significance of their result.

Crystal plasticity is certainly not independent of strain, since the manifestation of an anisotropic stress field is strain. Magmas ascending and extruding are mantled by shear zones in which strain localisation dominates, as a result of the shear stresses imposed – we aim to use crystal plasticity in the microlites to indicate where these areas are or were, their dominance in erupted material, and, since the magnitude of the plasticity suffered scales to the conditions imposed, we propose that plasticity can indicate the amount of deformation suffered. Given this, and thanks to the suggestion from reviewer 1, and the issue raised by reviewer 2, we have decided to refer to the deformed microlites as indicators of strain, though we wish to emphasise that we do believe this could be quantified systematically in future efforts to eventually serve as a strain marker in system-specific scenarios (i.e. for magma with known physical properties).

2. The two interpretations presented for the analytical data are difficult to understand. First, the data in Fig. 2 show large scattering. The authors simply used a linear fitting on the data. Using this fitting, they obtained an important parameter, the average misorientation per length; however, they completely neglected discussing the error in this parameter. As pointed out previously, the data show a large scatter; hence, the simple linear fit cannot be accepted without a quantification of error.

The line is not a FIT, we show in the supplementary information that the deformation scales more systematically with length than with aspect ratio, and as such, by taking the length and misorientation of each crystal we come up with a metric to compare intensity of deformation across crystals of different size – misorientation per micron. We now added a new figure (fig 6, shown right) which shows in a box-plot a statistical analysis of this values across each sample set, showing the min (bottom of the whisker), max (top of the whisker), median (horizontal in the centre of the box), 25th and 75th percentiles (bottom and top of the boxes, respectively) and the mean (the X) for each sample set. In this dataset we see the misorientation per micron for each sample set, which represents the lines in Figure 5 (previous fig. 2). So while the scatter is relatively high, as quantified in the new figure, there is no inherent error in this value, since it is given to allow a first order prediction of the amount of crystal plasticity for a crystal of a given length - possible because it is scalar value (unlike aspect ratio, for example it would not work).

The interpretation of brittle to plastic deformation transition of plagioclase microlite is also difficult to understand. The authors inferred that the plastic deformation results in brittle fracturing at a threshold for maximum lattice misorientation. However, the fractured microlite in Fig. 1E (central fragment) does not show misorientation.

I append the figure in point below (now fig. 4): please note that the microlite clearly exhibits plasticity across all fragments (gradual change in colour shading = change in orientation). We direct the reviewer to supplementary Table 10 to the broken crystals in sample B2, where they can find the

data for the misorientation profiles. This clearly shows the deformation within the middle fragment – I have also pasted this transect here for clarity, demonstrating that the 3.4 micron long crystal has >1.7degrees of misorientation – a misorientation per micron of 0.51 degrees/micron, almost exactly the median value for broken microlites in COLB2 (I suspect the reviewer has been misled by the vertical scale in panel F shown below).

In addition, in the Supporting Information (Figures S6–S8), many fractured crystals do not show misorientation. Without additional explanation, it appears that the authors are over-interpreting their data.

All microlites show internal crystal lattice distortion – plasticity. This point has clearly stemmed from a misunderstanding of the dataset by the reviewer. Of the figures to which the reviewers refer, Figure s6 is the phenocrysts (not microlites), the point being that these do not show crystal plasticity, and instead only show brittle fractures because they suffer higher stresses than the microlites by forming a rigid network in the samples, which forces them fully into the brittle regime. We state this in the manuscript (“We found the deformation of phenocrysts to be dominated by brittle fractures (Supplementary Figure S6) which often formed parallel to the principal stress direction (Supplementary Figure S3) and which resulted in a net grain size reduction during deformation”). Figures S7 and s8 show the Euler angles, they do not show whether there is or is not misorientation – we refer the reader to look supplementary tables (Tables S2-S10) to establish that every microlite is plastically deformed. From this comment, we recognise that there may be some confusion; as such, we have added a new figure now with band contrast images and pole figures for a EBSD map of the microlites from each sample (figure 3), and another which shows the misorientation per

micron across all samples, including the broken fragments (figure 6), showing the range of values measured and highlighting that all broken fragments suffered plasticity.

3. The relationship between deformation and misorientation is unclear because the experiments were performed using naturally deformed andesite. In a large undercooling system, lattice misorientation in clinopyroxene forms in the crystallization process, not deformation (e.g., Hammer et al., 2010). Brugger and Hammer (2015) reported that no misorientation was found in plagioclase that was formed in crystallization experiments. Based on the previous data, the authors need to discuss the formation of misorientation during crystallization. In addition, I strongly suggest that the authors compare the analytical results from natural samples with different degrees of deformation, i.e., non-deformed lava and shear-zone lava. All of these data support the interpretations presented in this study.

Deformation, not crystallisation is the focus here, as asserted experimentally. The natural samples that we use for the experiments already have a base level of microlite distortion, which we use as the comparison point here for our experiments where we take the same lava and deform it more under controlled conditions. By creating more plasticity during our experiments, in which the microlites are not crystallising but simply deforming (we show the material before and after, there are no more crystals, there is no evidence of growth, and indeed at the temperature and timescale of the experiments one would not expect crystal growth), we make it a moot point as to whether misorientation can result from crystallisation in plagioclase – here, it is a result of deformation. Further, the reviewer states “Brugger and Hammer (2015) reported that no misorientation was found in plagioclase that was formed in crystallization experiments.”, so indeed there is no evidence that this is the case in plagioclase at all and this reference supports our interpretation that misorientations observed in this study are due to deformation and are not growth (crystallisation) structures., We cannot comment on the behaviour of pyroxene (which might be very different from plagioclase) as this is not investigated in our study. Finally, we do not see that adding data examining variously sheared natural samples would add to the study, since we are here demonstrating that misorientation in the microlites scales to known, controlled conditions (those imposed in the experiments), detailing its potential use as an indicator of strain - it would be an entirely new endeavour to begin the process of examining variations in natural samples, and one which would rely upon the acceptance of the concept presented herein a priori.

4. It would be helpful to provide the details of the EBSD experimental design. The absolute orientations from EBSD measurements depend on the design and implementation of the experiment (e.g., Kilian et al., 2016). This is just a confirmation; however, the data reliability will be supported by the explanation.

We have added information on the reference frames that the Liverpool EBSD-SEM labs use during EBSD data collection and presentation in supplementary material. We are aware of the work of Kilian et al. (2016) and have tested our system thoroughly (see SI). We would like to point the reviewer to the work of Britton et al. (2016), where first-principle corrections of absolute orientations from EBSD measurements are reported. For consistency and clarity we have added the reference frame use to all our EBSD data (maps and pole figures) for clarity.

Other comments

Line 73: Please provide the chemical composition of the plagioclase phenocryst and microlite. The plagioclase present in the andesites recently erupted at Volcán de Colima range between andesine and labradorite. This has been subject of previous geochemical studies and we refer the readers to Reubi and Blundy (2008) and Savov et al. (2008), which we now cite in the manuscript.

Line 76: How did you determine the error in the pores? On the other hand, why do the solid portions have no error?

Porosity was measured multiple times (on 5 sample cores) so this is the range not error. Solid portions were measured once using the thin sections of the starting materials, as presented in Kendrick et al., 2013, hence there is no range. We realise that this information should have been expressed in the caption for Table 2 and have now added it.

Line 88: Have the authors measured the porosity and permeability? This data would also be interesting.

Yes the porosity was a few lines above (as shown in Table 2), the permeability is in Kendrick et al. 2013, which we refer to in terms of the rearrangement of void space, but it is not central to the story here.

Line 143: To identify the trend, an estimation of the error is necessary (see general comment 2). This is not a trend line, please see previous reply – we’ve tried to clarify in the manuscript how we came up with these lines, as evidently it was not explained clearly enough in the previous version.

Line 154: Figure 2 clearly indicates that the strain does not cause an increase in the misorientation. Why do the authors consider the misorientation as a strain maker? (see general comment 1).

The authors disagree with this comment, the strain must be active to allow the material to experience stress. Please refer back to the earlier relevant replies here, pasted below for convenience, and note that we have changed the terminology to “indicator of strain”.

Based on a couple of queries from both reviewers we decided to perform some further statistical analysis of our dataset. Using the metric “misorientation per micron” which is simply the lattice distortion (in degrees) divided by the crystal length (in microns) as documented in the manuscript, we investigated the maximum, minimum, median, and plotted these on a box-plot along with the 25th and 75th percentiles of the values. Using the misorientation per micron allows us to exclude any influence of measuring larger or smaller crystals in any sample. The box-plot shown to the right gives the results of the further analysis and demonstrates a number of important points. With regard to

the above question, we can see that the mean value (x on the plots) of misorientation per micron increases as deformation conditions increase (from left to right) in both samples, as does the minimum value and the maximum value – so, increasing strain the same stress does increase all 3 of these in a statistically relevant manner. Interestingly though, in both the samples the median, and both the 25th and 75th percentiles change very little from 20 to 30% strain at 28MPa, suggesting that while the deformation of some microlites certainly increases (the max, min and mean increase), some of the microlites may not – this could be due to strain partitioning coming into effect after a certain amount strain, such that not all microlites are still suffering deformation. This would be a really interesting avenue to explore, perhaps with in-situ synchrotron experiments, or with higher time-resolution

investigations (and perhaps a simpler system).

Line 169: Quantitative data and error estimation need to be provided to indicate the size reduction, i.e., the crystal size distribution should be measured and provided.

We already provided this information for each data-set in supplementary Tables S2-S10, where we show the length and area of each microlite measured. We also believe that the grain size reduction is clearly seen in panels A and B of Figure 5 (formerly Figure 2) in the main manuscript. However, we also now included grain size distribution as a plot in Supplementary Figure S7.

Line 198: Brittle fracturing does not seem to involve plastic deformation (see general comment 2).

Please see previous answer, unfortunately it seems the reviewer has confused the data between microlites (plastically deforming) and phenocrysts (brittle).

Line 214: 'stress/strain' is incorrect, because stress controls the formation of the misorientation while the misorientation is independent of strain. Stress and strain are different, and this expression is very confusing.

We concede to referring to "stress and strain", but believe that this statement is correct as per the answers regarding the significance of strain, particularly on the previous page, and given the new statistical information in figure 6.

Line 224: Misorientation cannot be used as a strain maker, although it depends on stress.

We have changed the phrase to "indicator of strain" but insist that it is indeed a function of strain, though stress is the driving force it is strain which results from differential stresses.

Line 234: How were the contents measured? How were the images obtained? SEM? Optical microscope?

This measurement refers again to the solid portion (%) data in Table 2, we apologise that we missed the reference from Kendrick et al 2013 where the method is presented, here we simply converted the componentry to a solid fraction.

Line 237: Please verify the crystallization during heating and deformation. The analyses for samples which were heated but not deformed are also necessary. Ideally, this data should be compared with data obtained from the deformation experiment, because the natural sample does not include the effect of heating.

Under the timescale (<1hr) tested at 945 °C, we observe no crystallisation or melting of the crystalline phases, as diffusion is very sluggish in a dry interstitial melt with such a high viscosity (estimated at $\sim 10^{8.58}$ Pa.s in Lavallée et al., 2007). We have asserted this through careful textural analysis of thin sections (using optical microscope and SEM imaging) before and after heat treatment in this study as well as in previous studies (Kendrick et al., 2013; Lavallée et al., 2007; 2008; 2012; 2013; 2017).

Line 252: Tilt is 70°?

This has now been clearly stated in the methods.

Supplementary Figures 7 & 8: Plagioclase microlite shows twinning. The formation mechanism of the twinning should be discussed.

The twins in the microlites are growth twins, but it is not overly abundant and as such we cannot see why the formation mechanisms of the twins is relevant to the study, especially since we always only

analyse one twin of each crystal to not bias the deformation data. Owing to its minor occurrence, we have not added a discussion of the twinning. There may be more appropriate volcanic products to study the impact of twinning on crystal plasticity and if we eventually come across this material, we will certainly consider delving further into this process.

References

- Brugger, C.R. and J.E. Hammer (2015) Prevalence of growth twins among anhedral plagioclase microlite. *American Mineralogist*, 100, 385–395.
- Cashman, K.V., C.R. Thornber and J.S. Pallister (2008) From dome to dust: Shallow crystallization and fragmentation of conduit magma during the 2004 – 2006 dome extrusion of Mount St. Helens, Washington.
- Dingwell, D.B. (1996) Volcanic dilemma: Flow or blow? *Science*, 273, 1054–1055.
- Hammer, J.E., T.G. Sharp and P. Wessel (2010) Heterogeneous nucleation and epitaxial crystal growth of magmatic minerals. *Geology*, 38, 367–370.
- Kilian, R., M. Bestmann and R. Heilbronner (2016) Absolute orientations from EBSD measurements - as easy as it seems? *Geophysical Research Abstracts*, 18, EGU2016-8221.
- Tuffen, H. and D. Dingwell (2005) Fault textures in volcanic conduits: evidence for seismic trigger mechanism during silicic eruptions. *Bulletin Volcanology*, 67, 370–387.

Additional references cited here:

- Cordonnier, B., Caricchi, L., Pistone, M., Castro, J., Hess, K. U., Gottschaller, S., Manga, M., Dingwell, D. B., and Burlini, L., 2012, The viscous-brittle transition of crystal-bearing silicic melt: Direct observation of magma rupture and healing: *Geology*, v. 40, no. 7, p. 611-614.
- Deer, W. A., Howie, R. A. & Zussman, J. *An Introduction to the Rock-Forming Minerals*. (2013).
- Kendrick, J. E., Lavallée, Y., Hess, K. U., Heap, M. J., Gaunt, H. E., Meredith, P. G., and Dingwell, D. B., 2013, Tracking the permeable porous network during strain-dependent magmatic flow: *Journal of Volcanology and Geothermal Research*, v. 260, p. 117-126.
- Lavallée, Y., Benson, P. M., Heap, M. J., Hess, K.-U., Flaws, A., Schillinger, B., Meredith, P. G., and Dingwell, D. B., 2013, Reconstructing magma failure and the degassing network of dome-building eruptions: *Geology*, v. 41, no. 4, p. 515-518.
- Lavallée, Y., Heap, M. J., Kueppers, U., Kendrick, J. E., and Dingwell, D. B., 2017, The fragility of Volcán de Colima – a material constraint, in Varley, N. R., and Komorowski, J.-C., eds., *Volcán de Colima – Managing the Threat*, Volume in press, Springer.
- Lavallée, Y., Hess, K.-U., Cordonnier, B., and Dingwell, D. B., 2007, Non-Newtonian rheological law for highly crystalline dome lavas: *Geology*, v. 35, no. 9, p. 843-846.
- Lavallée, Y., Meredith, P. G., Dingwell, D. B., Hess, K. U., Wassermann, J., Cordonnier, B., Gerik, A., and Kruhl, J. H., 2008, Seismogenic lavas and explosive eruption forecasting: *Nature*, v. 453, no. 7194, p. 507-510.
- Lavallée, Y., Varley, N. R., Alatorre-Ibarguengoitia, M. A., Hess, K. U., Kueppers, U., Mueller, S., Richard, D., Scheu, B., Spieler, O., and Dingwell, D. B., 2012, Magmatic architecture of dome-building eruptions at Volcan de Colima, Mexico: *Bulletin of Volcanology*, v. 74, no. 1, p. 249-260.
- Reubi, O., and Blundy, J., 2008, Assimilation of Plutonic Roots, Formation of High-K Exotic Melt Inclusions and Genesis of Andesitic Magmas at Volcán De Colima, Mexico: *Journal of Petrology*, v. 49, no. 12, p. 2221-2243.
- Savov, I. P., Luhr, J. F., and Navarro-Ochoa, C., 2008, Petrology and geochemistry of lava and ash erupted from Volcan Colima, Mexico, during 1998-2005: *Journal of Volcanology and Geothermal Research*, v. 174, no. 4, p. 241-256.

Reviewers' comments:

Reviewer #1 (Remarks to the Author):

I have reviewed the authors' response letter in detail. The authors should be congratulated on the completeness of their revisions and they have adequately addressed the comments raised by both reviewers.

Reviewer #2 (Remarks to the Author):

Review of "Crystal plasticity as an indicator of the viscous-brittle transition in magmas", by Kendrick et al.

In the revised version of the paper, the authors mostly addressed the reviewer comments. The methods and observations are acceptable at this time. However, four important points related to the interpretation of data and the discussion should be resolved before the paper is published.

First, the authors propose that the misorientation per microlite length can be used as a marker of strain, but no clear relationship between misorientation and strain was observed (new Fig. 6a). Only the maximum of the misorientation increases slightly with strain (the median, mean, and 25th and 75th percentages are independent of strain). This may indicate shear localisation at large strain. The authors should address this topic more carefully. Second, the presence of plastic deformation may be used for as a marker of the viscous-brittle transition of magma. The authors also emphasized, in their responses to the reviewer comments, that "crystal-plasticity is a necessary stage preceding failure". However, phenocrysts showed no plastic deformation, although they were strongly broken. If crystal-plasticity is a necessary stage, plastic deformation should also be observed in phenocrysts. The careful explanation for the distinction between phenocryst and microlite is necessary. A possible scenario is that phenocrysts make rigid framework and interstitial melt including microlite deforms, resulting in the breakage of phenocrysts and plastic deformation of microlite with/without brittle failure. If so, crystal-plasticity cannot be used as a maker of the viscous-brittle transition. Third, the relationship between the brittle failures of melt and crystal is unclear. In pure melts, the brittle failure of the melt is related to the ratio of the relaxation timescale of the melt to the deformation timescale (Dingwell, 1996). In contrast, the failure of crystal-bearing magma may be controlled by stress concentration on the melt due to the crystals (e.g., Ichihara and Rubin, 2010). The new Figure 7 indicates that the failures of melt and crystal have the same timing. A more careful explanation is necessary. Finally, the schematic figure (Fig. 7) includes a point that is difficult to understand. The velocity profile is shown by solid curves at the bottom of the conduit. However, in the central part of the conduit, no velocity profile is shown, which indicates no shear strain, although the solid arrow toward the surface indicates an increase in strain. Based on this consideration, the white curve representing the onset of crystal plasticity is also difficult to understand.

Minor comments

Lines 17–18: Change "increasing deformation" to "increasing stress".

Line 20: Broken phenocrysts do not show plastic deformation. Do the phenocrysts exceed the plastic limit?

Line 76: Please indicate the microlite composition (see the comments from reviewer #2).

Lines 164–165: Change "increasingly more deformed samples" to "samples deformed under large stress".

Lines 177–178: Only the maximum of the misorientation per length increases with strain. This

may indicate shear localisation at large strain. If so, the mean and median cannot be used as a marker of strain, but the maximum has potential as a strain marker.

Replies in black to reviewer comments in blue below.

Reviewer #1 (Remarks to the Author):

I have reviewed the authors' response letter in detail. The authors should be congratulated on the completeness of their revisions and they have adequately addressed the comments raised by both reviewers.

Reviewer #1 was very satisfied with the revisions and requested no further changes or clarifications.

Reviewer #2 (Remarks to the Author):

Review of “Crystal plasticity as an indicator of the viscous-brittle transition in magmas”, by Kendrick et al. In the revised version of the paper, the authors mostly addressed the reviewer comments. The methods and observations are acceptable at this time.

We are pleased to note that reviewer #2 finds the methods and observations to have been revised appropriately and is satisfied with these areas.

However, four important points related to the interpretation of data and the discussion should be resolved before the paper is published.

First, the authors propose that the misorientation per microlite length can be used as a marker of strain, but no clear relationship between misorientation and strain was observed (new Fig. 6a). Only the maximum of the misorientation increases slightly with strain (the median, mean, and 25th and 75th percentages are independent of strain). This may indicate shear localisation at large strain. The authors should address this topic more carefully.

This is not quite accurate, the minimum, maximum and mean all increase with increasing strain (see fig 6 and detailed description at lines 170-180 as well as all the values in supplementary Tables 2-10) – it is only the median, 25th and 75th percentiles which appear less sensitive to strain over the range measured, suggesting that at higher strains there is a more pronounced effect of strain localisation (as we detail in the text, and we note that strain localisation at higher strains has been observed in other studies, cited in the text) – i.e. some crystals are deformed substantially more than others, the least deformed microlites remain at a similar level of deformation as the lower strain sample. A final comment regarding the use as a strain marker: Given that strain is the deformation resulting from a stress, it follows that all the microlites examined here are subject to strain of differing amounts – crystal plasticity itself is otherwise referred to as plastic strain – hence, we firmly believe that quantifying crystal plasticity serves to quantify local strain (if not bulk system strain, though this may in future be possible in well-constrained systems) as well as more broadly offering a tool to map strain localisation. We have reworded this section (lines 168-188) as carefully as possible to try to be as clear as possible about what our description of strain is, what the data here suggest and what the reason behind that might be, in light of the reviewer’s comments (in this and the previous round of reviews) and we hope that this will be satisfactory.

Second, the presence of plastic deformation may be used for as a marker of the viscous-brittle transition of magma. The authors also emphasized, in their responses to the reviewer comments, that “crystal-plasticity is a necessary stage preceding failure”. However, phenocrysts showed no plastic deformation, although they were strongly broken. If crystal-

plasticity is a necessary stage, plastic deformation should also be observed in phenocrysts. The careful explanation for the distinction between phenocryst and microlite is necessary. A possible scenario is that phenocrysts make rigid framework and interstitial melt including microlite deforms, resulting in the breakage of phenocrysts and plastic deformation of microlite with/without brittle failure. If so, crystal-plasticity cannot be used as a maker of the viscous-brittle transition.

The reviewer points out that we may have overstated our claim in the previous replies, that “plasticity always precedes failure” – we must clarify that this was in reference to the system described – i.e. the microlites in the Colima andesite, but re-reading the comment we realised it was not clear this was in discussion of the microlites herein, and apologise for this. We believe the manuscript offered a fairer appraisal, for example, in the conclusion we state “Further, crystal plasticity necessarily complicates the viscous-brittle transition envisaged during magma ascent (Cordonnier et al., 2012; Dingwell, 1997; Kendrick et al., 2013b; Lavallée et al., 2012; Pistone et al., 2015; Shields et al., 2014), providing a time-space interval during which strain may be accommodated by crystal plastic deformation (Fig. 7).” We have now clearly expressed that, if material is deformed sufficiently rapidly, or at sufficiently high load, then one would expect the material to behave in a purely brittle manner – this stands for crystals, for melt or any other material - we believe this to be the case in the phenocrysts, which here are able to form a rigid network within the deforming magma, concentrating stress and breaking. There is a great simulation of this stress build-up in Deubelbeiss et al. (2011). We contest that this invalidates plasticity as a marker of the viscous brittle transition in magmas, since all viscous systems such as that described here will have both phenocrysts and microlites, and the authors believe crystal plasticity to be important in the microlites of all such eruptions. [It is worthwhile noting that liquids themselves may plastically deform in the glass transition interval (e.g. Le Bourhis, 2014), thus plasticity is a phenomenon that deserves further consideration in volcanology.] Further, as we propose in the discussion, we also believe crystal plasticity is relevant at other rheological conditions (temperatures- strains- strain rates) in magmatic systems, as has been demonstrated in work on intrusive magma bodies (see e.g. Webber et al., 2015). We hope the reviewer will be satisfied with the changes in lines 104-117 that better detail the development of fractures in the phenocrysts, and lines 242-245 in the discussion as well as numerous locations throughout the manuscript (see tracked changes).

Third, the relationship between the brittle failures of melt and crystal is unclear. In pure melts, the brittle failure of the melt is related to the ratio of the relaxation timescale of the melt to the deformation timescale (Dingwell, 1996). In contrast, the failure of crystal-bearing magma may be controlled by stress concentration on the melt due to the crystals (e.g., Ichihara and Rubin, 2010). The new Figure 7 indicates that the failures of melt and crystal have the same timing. A more careful explanation is necessary.

The reviewer raises a good point, but we do not attempt to tackle this complex issue in the present study. We do not mean to suggest that fractures form simultaneously in crystals and melt, figure 7 is simply a schematic, with no time-series indicated – between area 2 and area 1 in the schematic, fractures have formed. In natural systems, damaged crystals often serve as our only relic of deformation, since the melt is able to relax, and heal/sinter fractures back together if provided with sufficient timescales (e.g. Vasseur et al., 2013; Wadsworth et al., 2016), so establishing the temporal relationship of fracturing is tricky. In actual fact, where the fractures form first will be different depending on the melt viscosity, crystallinity (and size/shape distributions) and porosity – Vasseur et al. (2015) elegantly demonstrated that heterogeneities are the key to failure forecasting, in other words, that heterogeneities (i.e. pores or crystals) originate cracking events, while a few experimental studies on 3-phase

magmas (e.g. Cordonnier et al., 2009; Lavallée et al., 2007; Lavallée et al., 2012) have shown that fractures originate within the phenocrysts which formed a rigid network in relatively highly crystalline systems (again I point to the simulation of Deubelbeiss et al. (2011) for a great indication of how disparate stress build-up can be in suspensions compared to the bulk stress). In Cordonnier et al. (2012) and Lavallée et al. (2007) they show that these fractures were able to propagate into the melt – presumably due to excursions of the melt into the brittle field as a result of strain rate fluctuation (see for example Ichihara and Rubin (2010) which the reviewer recommends) as the crystal finally yields under stress, hence cracks can form more-or-less simultaneously in crystals and melt, but originating from the crystals. The photomicrographs in figure 2 here certainly show fractures through the phenocrysts and groundmass (glass and microlites) alike, though the phenocrysts appear to damage prior to full coalescence of through-going fractures (as we say in the caption). We added a brief discussion of the above points at lines 108-113 in the manuscript to ensure the topic was addressed, but don't feel that the schematic mis-represents our point.

Finally, the schematic figure (Fig. 7) includes a point that is difficult to understand. The velocity profile is shown by solid curves at the bottom of the conduit. However, in the central part of the conduit, no velocity profile is shown, which indicates no shear strain, although the solid arrow toward the surface indicates an increase in strain. Based on this consideration, the white curve representing the onset of crystal plasticity is also difficult to understand.

The schematic shows that 1) there is a velocity profile across the conduit which approximates plug-flow (and shear stress is higher at the conduit margins), and 2) total strain increases during ascent; we do not show (or model) how this stress profile evolves along the conduit, since it is not-to-scale but is supposed to insinuate that there remains a velocity profile throughout. There is not sufficient space to clearly show this throughout the conduit; it's simply a sketch and there's a lot of other detail we wish to highlight; here, the dashed white line that marks the onset of plasticity attempts to show that there is a stress-strain rheological point beyond which crystals (here microlites) can deform plastically (which would be subject to fluctuations due to ascent rate). We edited the caption to try to express that the velocity profile (hence stress and strain) are active along the length of the conduit, and hope this explains sufficiently.

Minor comments

Lines 17–18: Change “increasing deformation” to “increasing stress”.

We have changed the sentence, which no longer states “increasing deformation”, and the abstract is now more in-line with reviewer #2's thesis of the work presented (as per the main comments above).

Line 20: Broken phenocrysts do not show plastic deformation. Do the phenocrysts exceed the plastic limit?

As we say above, and as we have now made clear in the manuscript – the phenocrysts are not subject to the plastic limit, since this is a rate-dependent phenomena. If the material is deformed sufficiently rapidly, or at sufficiently high load, then one would expect the materials to behave in a purely brittle manner (e.g. Ichihara and Rubin, 2010 and many others in material science) – we believe this to be the case in the phenocrysts, which here are able to form a rigid network within the deforming magma, concentrating stress and breaking, as has been previously shown in multi-phase magmas (e.g. Cordonnier et al., 2009; Kendrick et al., 2013a; Lavallée et al., 2007; Lavallée et al., 2012).

Line 76: Please indicate the microlite composition (see the comments from reviewer #2).

The composition of the plagioclase (andesine to labradorite) is given along with the references cited (Reubi et al., 2013; Savov et al., 2008); yet it is not pivotal to this study. We have verified that the microlites in our samples have the same composition, using microprobe (see table below), which shows (given that Labradorite is $\text{Na}_{0.4}\text{Ca}_{0.6}\text{Al}_{1.6}\text{Si}_{2.4}\text{O}_8$ and Andesine is $\text{Na}_{0.6}\text{Ca}_{0.4}\text{Al}_{1.4}\text{Si}_{2.6}\text{O}_8$) the plagioclase here would be $\text{Na}_{0.47}\text{Ca}_{0.53}$, and hence the An:Ab ratio is almost exactly 50:50 on average from 10 points. While we trust this data's accuracy to verify previously published quantification of plagioclase composition, the measurements were unfortunately made during a change-over of the maintenance personnel in Munich and the measurement parameters were not recorded, so we do not wish to publish this data.

COLB2														
	Na	Mg	Al	Si	Ba	P	K	Ca	Ti	Cr	Mn	Fe	O	Total
1	3.8211	0.0386	10.6978	19.5663	0.0045	0.0133	0.0724	4.1173	0.0109	0.0034	0.0091	0.1448	61.5005	100
2	3.5306	0.0262	11.0895	19.2517	0.0026	-0.0044	0.0718	4.3822	0.0082	0.0006	0.0005	0.1421	61.4985	100
3	3.884	0.0363	10.5513	19.7974	0.0019	0.0122	0.0809	3.916	0.0068	-0.0039	0.0016	0.159	61.5568	100
4	3.6848	0.0127	10.7655	19.6978	0.0063	-0.007	0.0911	3.9795	0.0107	0.0045	-0.0048	0.1612	61.5975	100
COLLAH4														
	Na	Mg	Al	Si	Ba	P	K	Ca	Ti	Cr	Mn	Fe	O	Total
1	3.7304	0.0319	10.5784	19.8059	0.0024	-0.0019	0.0985	3.987	0.016	-0.003	-0.0068	0.1652	61.5962	100
2	3.6768	0.036	10.8162	19.505	0.0045	0.0079	0.0847	4.1667	0.0059	-0.0125	0.0128	0.174	61.5219	100
3	3.572	0.024	11.0376	19.2918	0.0029	0.0018	0.0748	4.2924	0.0091	0.0017	0.0005	0.1915	61.4999	100
4	4.0792	0.0328	10.4399	19.8932	0.0002	-0.0062	0.0858	3.7705	0.0139	0.0023	0.008	0.1622	61.5182	100
5	4.0469	0.0142	10.6756	19.4951	-0.0017	0.0054	0.0712	4.0894	0.0062	0.0041	0.0169	0.1816	61.3951	100
6	3.2479	0.0179	11.1168	19.1546	-0.0001	0.0062	0.0688	4.6235	0.013	0.0057	0.0054	0.2018	61.5404	100

Lines 164–165: Change “increasingly more deformed samples” to “samples deformed under large stress”.

We have changed this sentence as suggested.

Lines 177–178: Only the maximum of the misorientation per length increases with strain.

This may indicate shear localisation at large strain. If so, the mean and median cannot be used as a marker of strain, but the maximum has potential as a strain marker.

This comes back to main point 1 of reviewer #2, to which I refer the reviewer to the above reply (and additionally noting that we do not refer to crystal plasticity as a strain marker any more, we point to its potential as such in the abstract and discussion, but primarily now focus on describing it as a tool to quantify deformation, especially across similar systems deformed under contrasting conditions – we think that the review process has helped to significantly improve both the clarity and message of the manuscript in this respect).

References cited

- Cordonnier, B., Caricchi, L., Pistone, M., Castro, J., Hess, K.-U., Gottschaller, S., Manga, M., Dingwell, D. B., and Burlini, L., 2012, The viscous-brittle transition of crystal-bearing silicic melt: Direct observation of magma rupture and healing: *Geology*, v. 40, no. 7, p. 611-614.
- Cordonnier, B., Hess, K.-U., Lavallée, Y., and Dingwell, D. B., 2009, Rheological properties of dome lavas: Case study of Unzen volcano: *Earth and Planetary Science Letters*, v. 279, no. 3-4, p. 263-272.
- Deubelbeiss, Y., Kaus, B. J. P., Connolly, J. A. D., and Caricchi, L., 2011, Potential causes for the non-Newtonian rheology of crystal-bearing magmas: *Geochem. Geophys. Geosyst.*, v. 12, no. 5, p. Q05007.
- Dingwell, D. B., 1997, The Brittle–Ductile Transition in High-Level Granitic Magmas: Material Constraints: *Journal of Petrology*, v. 38, no. 12, p. 1635-1644.

- Ichihara, M., and Rubin, M. B., 2010, Brittleness of fracture in flowing magma: *Journal of Geophysical Research: Solid Earth*, v. 115, no. B12, p. n/a-n/a.
- Kendrick, J. E., Lavallée, Y., Hess, K.-U., Heap, M. J., Gaunt, H. E., Meredith, P. G., and Dingwell, D. B., 2013a, Tracking the permeable porous network during strain-dependent magmatic flow: *Journal of Volcanology and Geothermal Research*, no. 0.
- Kendrick, J. E., Lavallée, Y., Hess, K. U., Heap, M. J., Gaunt, H. E., Meredith, P. G., and Dingwell, D. B., 2013b, Tracking the permeable porous network during strain-dependent magmatic flow: *Journal of Volcanology and Geothermal Research*, v. 260, p. 117-126.
- Lavallée, Y., Hess, K. U., Cordonnier, B., and Dingwell, D. B., 2007, Non-Newtonian rheological law for highly crystalline dome lavas: *Geology*, v. 35, no. 9, p. 843-846.
- Lavallée, Y., Varley, N., Alatorre-Ibargüengoitia, M., Hess, K. U., Kueppers, U., Mueller, S., Richard, D., Scheu, B., Spieler, O., and Dingwell, D., 2012, Magmatic architecture of dome-building eruptions at Volcán de Colima, Mexico: *Bulletin of Volcanology*, v. 74, no. 1, p. 249-260.
- Le Bourhis, E., 2014, *Glass: Mechanics and Technology*, 2nd Edition, Wiley.
- Pistone, M., Cordonnier, B., Caricchi, L., Ulmer, P., and Marone, F., 2015, The viscous to brittle transition in crystal- and bubble-bearing magmas: *Frontiers in Earth Science*, v. 3.
- Reubi, O., Blundy, J., and Varley, N., 2013, Volatiles contents, degassing and crystallisation of intermediate magmas at Volcan de Colima, Mexico, inferred from melt inclusions: *Contributions to Mineralogy and Petrology*, v. 165, no. 6, p. 1087-1106.
- Savov, I. P., Luhr, J. F., and Navarro-Ochoa, C., 2008, Petrology and geochemistry of lava and ash erupted from Volca[combining acute accent]n Colima, Mexico, during 1998-2005: *Journal of Volcanology and Geothermal Research*, v. 174, no. 4, p. 241-256.
- Shields, J. K., Mader, H. M., Pistone, M., Caricchi, L., Floess, D., and Putlitz, B., 2014, Strain-induced outgassing of three-phase magmas during simple shear: *Journal of Geophysical Research: Solid Earth*, v. 119, no. 9, p. 6936-6957.
- Vasseur, J., Wadsworth, F. B., Lavallée, Y., Bell, A. F., Main, I. G., and Dingwell, D. B., 2015, Heterogeneity: The key to failure forecasting, v. 5, p. 13259.
- Webber, J. R., Klepeis, K. A., Webb, L. E., Cembrano, J., Morata, D., Mora-Klepeis, G., and Arancibia, G., 2015, Deformation and magma transport in a crystallizing plutonic complex, Coastal Batholith, central Chile: *Geosphere*.